# Descending inhibitory rostral ventromedial medulla neurons cause widespread antinociception and contribute to the pain-inhibits-pain phenomenon

Robert P. Ganley [1,2,8], Marília Sousa [1,3,8], Guangchen Ji[4], Matteo Ranucci [1,3], Camilla Beccarini [1,3], Kira Werder [1], Francesca Pietrafesa [1,3], Simon d'Aquin[1], Tugce Akyüz[5], Michèle Hubli[6], Petra Schweinhardt[7], Volker Neugebauer[4], Mark A. Hoon [2], Hendrik Wildner[1] & Hanns Ulrich Zeilhofer [1,3,5] ✉

Acute painful stimuli applied to one body site reduce pain at other sites. The circuit basis of this "pain-inhibits-pain" phenomenon, also known as diffuse noxious inhibitory control (DNIC) in animals or conditioned pain modulation (CPM) in humans, is largely unknown. Using anatomical and optogenetic circuit tracing, we identified a population of descending inhibitory neurons of the rostral ventromedial medulla (RVM) that densely and bilaterally innervate the spinal cord along its rostrocaudal axis. Activating these neurons reduced heat and cold sensitivity widely in healthy mice and caused similarly wide-spread antihyperalgesia in chronic pain models, while their silencing evoked mechanical allodynia and spontaneous pain-like behaviors. Noxious stimuli activated subsets of these neurons in the lateral paragigantocellularis nucleus (LPGi), which inhibited nociception upon chemogenetic reactivation. Spinally projecting inhibitory RVM neurons are hence ideally positioned to function as circuit elements of DNIC and CPM, while their dysfunction may contribute to wide-spread chronic pain syndromes.

The rostral ventromedial medulla (RVM), which includes the nucleus raphe magnus, the nucleus paragigantocellularis pars alpha (Pa) and the lateral paragigantocellularis nucleus (LPGi), serves as a critical hub for descending pain modulation systems[1,2]. Well-known examples of endogenous context dependent pain modulation include among others stress-induced analgesia and the pain-inhibits-pain phenomenon. Descending projections from the RVM are particularly important for context-dependent changes in conscious pain perception and can tune pain sensitivity in a bidirectional manner[3,4]. The RVM is the origin of several distinct descending fiber tracts that differ in their neurochemistry and anatomical organization. Previous work has identified neurotransmitters and neuromodulators released from descending RVM neurons, including glutamate, GABA and/or glycine, and serotonin, that are potentially involved in descending pain modulation. However, the precise anatomical organization of the underlying circuits and projections, and their contribution to specific forms of endogenous pain modulation are still incompletely understood[5–8] (for recent reviews see ref. 9,10).

[1]Institute for Pharmacology and Toxicology, University of Zürich, Zürich, Switzerland. [2]Molecular Genetics Section, National Institute of Dentofacial and Craniofacial Research, NIH, Bethesda, MD, USA. [3]Neuroscience Center Zurich (ZNZ), University of Zürich, Zürich, Switzerland. [4]Department of Pharmacology & Neuroscience, Texas Tech University Health Sciences Center, Lubbock, TX, USA. [5]Institute of Pharmaceutical Sciences, ETH Zürich, Zürich, Switzerland. [6]Spinal Cord Injury Center, Balgrist University Hospital, University of Zurich, Zurich, Switzerland. [7]Department of Chiropractic Medicine, Balgrist University Hospital, University of Zurich, Zurich, Switzerland. [8]These authors contributed equally: Robert P. Ganley, Marília Sousa. ✉e-mail: zeilhofer@pharma.uzh.ch

It is well established that inhibition at the level of the spinal cord is mediated by the fast inhibitory neurotransmitters GABA and glycine, which tune nociceptive sensitivity to a physiologically meaningful level[11–13]. In addition to local spinal interneurons, supraspinal projections (such as those from the RVM) provide a source of GABA and glycine to the spinal dorsal horn[3,14]. We and others have recently shown that the superficial layers of the dorsal horn, i.e., the sites where nociceptive afferent fibers terminate, are densely innervated by GABAergic and glycinergic axons originating from the RVM[3,14,15]. It is hence conceivable that these projections contribute to context-dependent changes in pain sensitivity.

Here, we used intersectional approaches for high-precision anatomical and optogenetic circuit tracing to specifically investigate the spinal projections of inhibitory RVM neurons (vGAT RVM$^{SC}$ neurons). This approach was combined with chemogenetic and optogenetic activation and tetanus toxin mediated silencing to examine the biological functions of vGAT RVM$^{SC}$ neurons. We found that activating this circuit produced nearly body-wide, antinociception and antihyperalgesia via wide-ranging axons that project bilaterally throughout the entire length of the spinal cord. Experiments employing the Targeted Recombination in Active Populations (TRAP) technique demonstrated that a specific subset of these neurons, located mainly in the LPGi, were not only activated by noxious stimuli but also provided wide-spread analgesia upon chemogenetic reactivation. These projections therefore appear as critical elements of a circuit underlying the so-called "pain-inhibits-pain" phenomenon, also known as diffuse noxious inhibitory control (DNIC) in rodents and conditioned pain modulation (CPM) in humans. Accumulating evidence suggests that this control system is impaired in chronic pain conditions[16], and interventions to restore it may have therapeutic potential. Accordingly, our silencing experiments indicate that compromised vGAT RVM$^{SC}$ neuron function causes pronounced mechanical sensitization to dynamic and punctate stimuli.

## Results

### Specific labeling of descending inhibitory RVM projections using AAV2retro vectors in vGAT$^{cre}$ mice

To identify and manipulate neural pathways that descend from the RVM to the spinal cord, we used AAV2retro vectors to transduce RVM neurons via their axon terminals in the spinal dorsal horn[17]. Since some projection neurons are resistant to retrograde transduction with these vectors[18–20], we first tested whether AAV2retro vectors would efficiently label the descending RVM neurons. We injected AAV2retro.eGFP into the left lumbar dorsal horn of 9–10-week-old mice and prepared tissue containing the RVM for multiplex fluorescent in situ hybridization (mFISH) (Fig. 1A). We found that many eGFP-labeled cells contained vGAT (19.4 ± 4.1%) or vGluT2 (55.1 ± 7.0%) mRNA, indicating that both excitatory and inhibitory neurons can be targeted with these vectors. Because the inhibitory neurons provide particularly dense innervation to the superficial dorsal horn[3,14,15], we focused our analyses on this population. To specifically label these projections, we repeated the experiments in vGAT$^{cre}$ mice and with a cre-dependent eGFP reporter construct. Most eGFP$^+$ neurons in the RVM contained vGAT mRNA (83.6 ± 9.3%) with only few containing detectable vGluT2 (2.0 ± 2.0%), confirming eutrophic cre expression (Fig. 1B).

Since vGAT$^{cre}$ is expressed not only in RVM projection neurons but also in interneurons of the RVM and spinal cord, we used an intersectional approach to specifically target those inhibitory RVM neurons that descend to the spinal cord[21,22]. To this end, we injected the lumbar spinal cords of vGAT$^{cre}$ animals with AAV2retro vectors containing an optimized cre-dependent flippase (Flpo). One week later, the RVM was injected with a Flpo-dependent eGFP reporter construct (AAV9.dFRT.eGFP) (Fig. 1C). To assess the accuracy and transduction efficiency of the RVM injections with AAV9.dFRT.eGFP, an mCherry sequence was included in the AAV2retro (AAV2retro.flex.FLPo.mCherry) to directly

label projections. We found that with single midline injections it was not possible to efficiently label vGAT RVM$^{SC}$ neurons located in the lateral and ventral RVM, such as the LPGi (Supplementary Fig. 1). We therefore switched to bilateral injections (−5.8, ±0.5, 5.9 from Bregma), which reached more than 90% of the retrogradely labeled RVM projection neurons (Fig. 1C). This labeling strategy hence provided a reliable means of accessing most descending vGAT neurons across the RVM.

### Global wide-ranging projections of vGAT RVM$^{SC}$ neurons

The spinal dorsal horn has a highly structured laminar organization, which is reflected by the segregated innervation of laminae by sensory fibers relaying different sensory modalities[23,24]. We therefore characterized the regional innervation pattern of the spinal cord by vGAT RVM$^{SC}$ neurons (Fig. 2A–C). We found that the axons of these neurons densely innervate the superficial laminae with only sparse innervation of the rest of the spinal cord (Fig. 2B). Strikingly, while labeling was performed from a single unilateral spinal cord injection site (left lumbar enlargement), labeled axon terminals were found bilaterally and throughout all spinal segments and including the spinal trigeminal nucleus of the brainstem (SPV) (Fig. 2B, C). Labeled axon terminals were also present in additional supraspinal sites (Fig. 2C), including the nucleus of the solitary tract (NTS), the lateral parabrachial area (LPb), and the periaqueductal gray matter (PAG). Notably, many of these areas are part of the ascending pain pathways[25,26]. To better visualize this projection pattern in its entirety, we used CLARITY-based tissue clearing and light sheet microscopy[27]. These analyses revealed axons emanating from the RVM in both rostral and caudal directions and on both ipsilateral and contralateral sides (Supplementary Video 1).

To quantify the extent to which vGAT RVM$^{SC}$ neurons projected to both sides of the spinal cord, we injected AAV2retro vectors containing cre-dependent mCherry into the left dorsal horn and cre-dependent eGFP into the right dorsal horn of the lumbar spinal cord (Fig. 2D). Although local spread of the AAVs was restricted to the injected side, many retrogradely labeled vGAT RVM$^{sc}$ neurons (31.0 ± 1.5%) expressed both mCherry and eGFP, indicating that they projected bilaterally (Fig. 2D–F). Of those cell bodies that were labeled with only one fluorophore, most were found on the ipsilateral side (60.2 ± 4.6%) (Fig. 2F). We also quantified the extent to which vGAT RVM$^{SC}$ project to different spinal cord segments. To this end, we retrogradely labeled vGAT$^{cre}$ neurons from the cervical and lumbar spinal cords with different fluorophores (Fig. 2G). AAV2retro.flex.eGFP was injected into the lumbar cord and AAV2retro.flex.mCherry into the cervical cord. Many cells (39.7 ± 2.2%) were labeled with both eGFP and mCherry, indicating that at least about 40% of the neurons provided bilateral inhibition over many spinal segments. Such wide-spread inhibition is further supported by the results of our optogenetic and electrophysiological experiments shown below. The remaining neurons were labeled with either eGFP (33.7 ± 2.9%) or mCherry only (26.7 ± 4.5%) (Fig. 2H, I). An incomplete viral transfection efficacy may contribute to this finding. However, the behavioral data shown below that were obtained after tetanus toxin light chain-mediated silencing of vGAT RVM$^{SC}$ neurons support that the RVM also harbors descending vGAT neurons that provide more localized inhibition to the spinal dorsal horn.

### Chemogenetic activation of descending inhibitory RVM projection neurons reduces nociceptive sensitivity

Our tracing experiments in vGAT$^{cre}$ mice revealed dense innervation of the superficial dorsal horn, the first node in the ascending pain pathway. To assess the impact of these projections on nociception, we first performed chemogenetic activation experiments, introducing hM3Dq to the vGAT RVM$^{SC}$ neurons labeled from a unilateral site of the lumbar spinal cord using our intersectional strategy (Fig. 3A). The expression pattern of hM3Dq-mCherry in the spinal cord was bilateral (Fig. 3B)

and virtually identical to that found in our previous labeling experiments. Upregulation of the activity-dependent marker c-Fos in retrogradely labeled RVM neurons verified the efficacy of the chemogenetic approach (Fig. 3C). Activation of these projections reduced the sensitivity of the ipsilateral hindpaw to heat and cold stimuli (Fig. 3D). In addition, activation reduced significantly the sensitivity to von Frey mechanical and pin-prick stimulation, indicating that they were

capable of suppressing sensitivity to multiple sensory modalities. These changes in nociceptive sensitivity were not accompanied by detectable changes in responses to brush stimulation or in sensorimotor coordination assessed in the rotarod test. To validate that CNO-mediated effects depended on hM3Dq activation, we repeated the sensory assays in which mice had shown altered responses and compared the effects of CNO between animals expressing hM3Dq or, as a

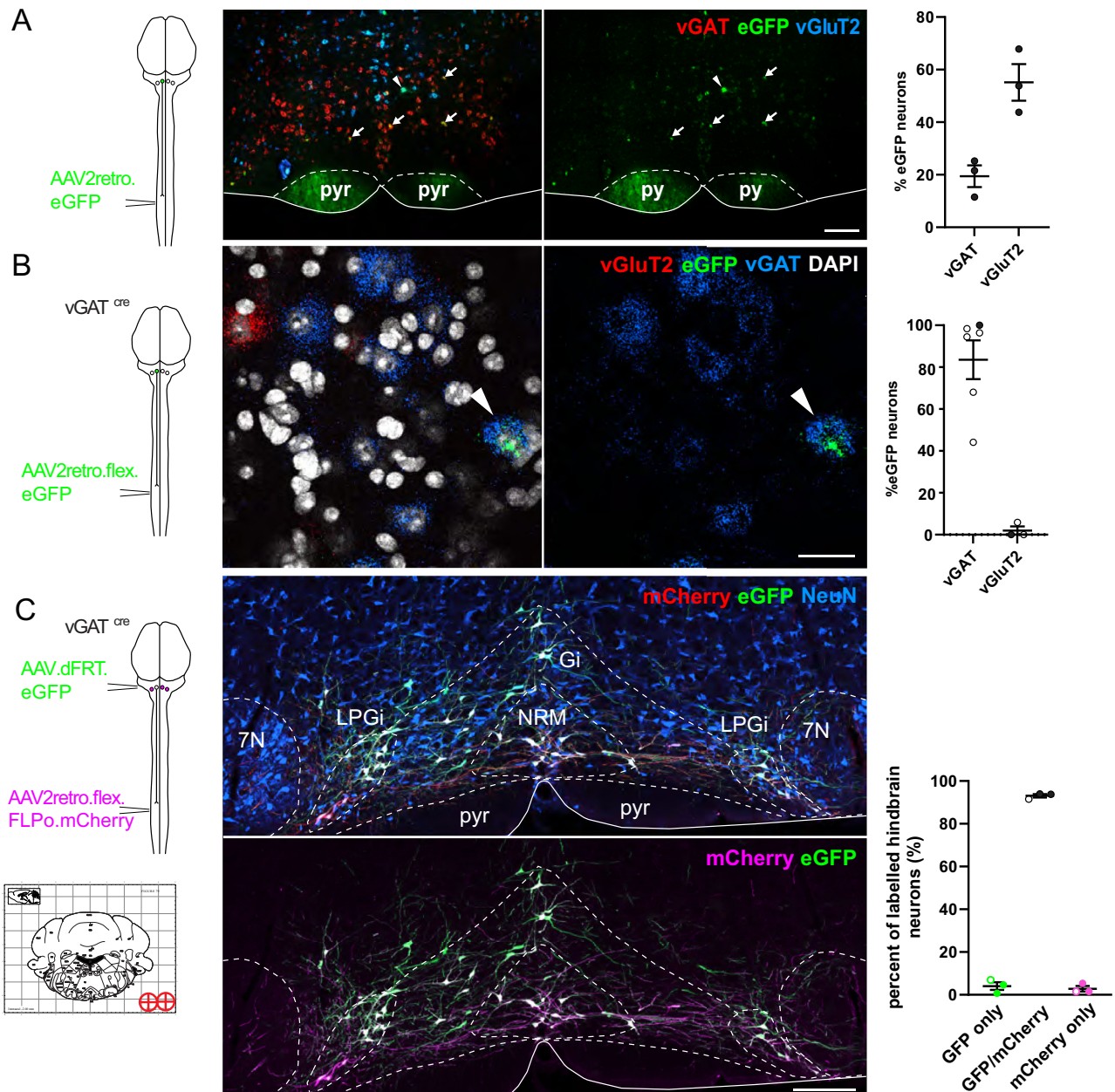

**Fig. 1 | Intersectional strategy for targeting inhibitory descending projection neurons. A** Left, injection scheme for labeling descending projection neurons with AAV2retro. Middle: micrographs showing mFISH of an RVM section with eGFP+ neurons that express vGAT mRNA (red, arrows) or vGluT2 mRNA (blue, arrowheads) (scale bar, 200 μm). pyr, pyramidal tract. Note that with this cre-independent labeling strategy all descending pathways were labeled including the corticospinal tract, axons of which run through the pyramids (pyr). Right: quantification of eGFP+ neurons that only colocalized with vGAT mRNA or vGluT2 mRNA. Each data point represents one mouse. Three sections per mouse were analyzed. **B** Left, injection scheme for the specific labeling of inhibitory neurons descending to the spinal dorsal horn. Middle, mFISH of RVM sections taken from vGAT^cre mice that received an intraspinal injection of AAV2retro.flex.eGFP (scale bar, 10 μm). The

arrowhead indicates an inhibitory projection neuron. Right: quantification of eGFP+ cells in the RVM containing vGAT or vGluT2 mRNA. Each data point represents one mouse (n = 6 mice (vGAT), n = 3 mice (vGluT2), and two to five sections were used per mouse. **C** Top left: intersectional strategy for the specific targeting of vGAT RVM^SC neurons. Top bottom: brain injection coordinates for the RVM injections and representative injection sites from labeling experiments (scale bar, 200 μm in both images). Middle: high magnification image of mCherry+ and eGFP+ labeled neurons in the RVM (scale bars, 100 μm). Right: quantification (mean ± SEM) of mCherry and/or eGFP expressing neurons in the RVM of vGAT^cre animals (n = 3). Open and closed symbols indicate female and male mice, respectively. Source data are provided as a Source Data file.

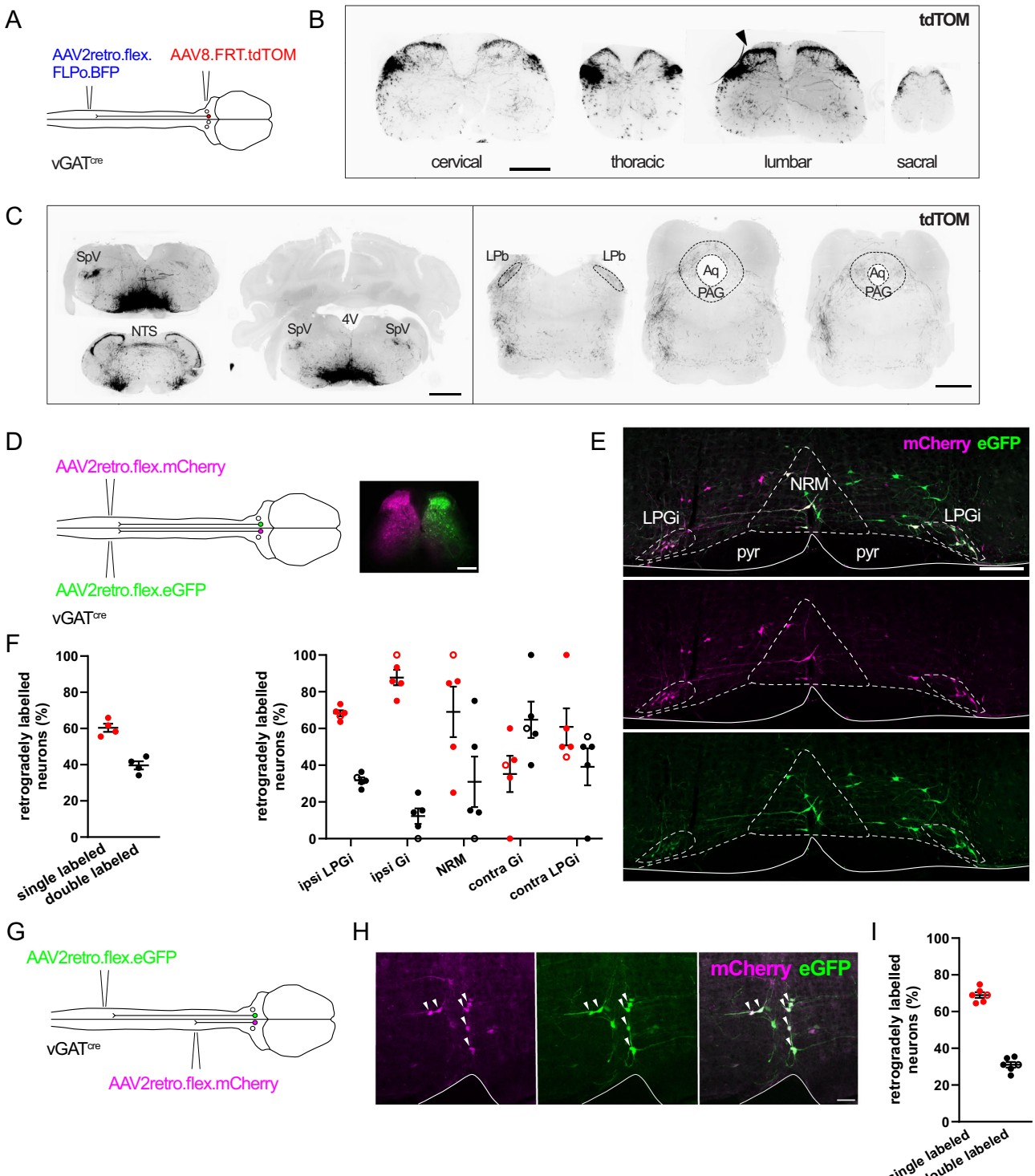

**Fig. 2 | Anatomical tracing of descending inhibitory RVM neurons. A** Injection scheme for labeling of vGAT RVM$^{sc}$ neurons with tdTOM. **B** Spinal cord sections from different segments illustrating the axon distribution of vGAT RVM$^{SC}$neurons. Arrowhead indicates the injection site of the AAV2retro.flex.FLPo.BFP into the left lumbar spinal cord (scale bar, 200 μm). **C** RVM injection site and rostral branches arising from vGAT RVM$^{SC}$ neurons. Axons are found in the spinal trigeminal nucleus, the LPb and PAG (scale bars, 500 μm). SpV spinal trigeminal nucleus, 4 V fourth ventricle, NTS nucleus of the solitary tract, LPb lateral parabrachial area, PAG periaqueductal gray matter, Aq aqueduct. Similar results were seen in n = 3 mice. **D** Injection scheme for retrogradely labeling RVM projection neurons from the left and right spinal cord of vGAT$^{cre}$ mice. Representative injection sites in a mouse (scale bar, 200 μm). **E** RVM neurons labeled from the injections illustrated in D (scale bar, 200 μm). NRM, nucleus raphe magnus; LPGi, lateral

paragigantocellularis; Pyr, pyramidal tract. Micrographs in D and E are taken from a single animal and are representative of n = 6 mice. **F** Quantification of vGAT RVM$^{SC}$ neurons labeled with either one (red circles) or two fluorophores (black circles) in different parts of the RVM (n = 6 mice) (data is displayed as mean ± SEM).
**G** Injection scheme for retrograde labeling of vGAT RVM$^{SC}$ neurons from different spinal segments (cervical and lumbar). **H** Confocal micrographs showing the co-expression of mCherry and eGFP within many neurons (arrowheads).
**I** Quantification of RVM neurons retrogradely labeled with one (red circlaes) or both fluorophores (black circles) from the experiment depicted in G (scale bar, 50 μm) (n = 4 mice). Open and closed symbols indicate female and male mice, respectively. Data are displayed as mean ± SEM. Source data are provided as a Source Data file.

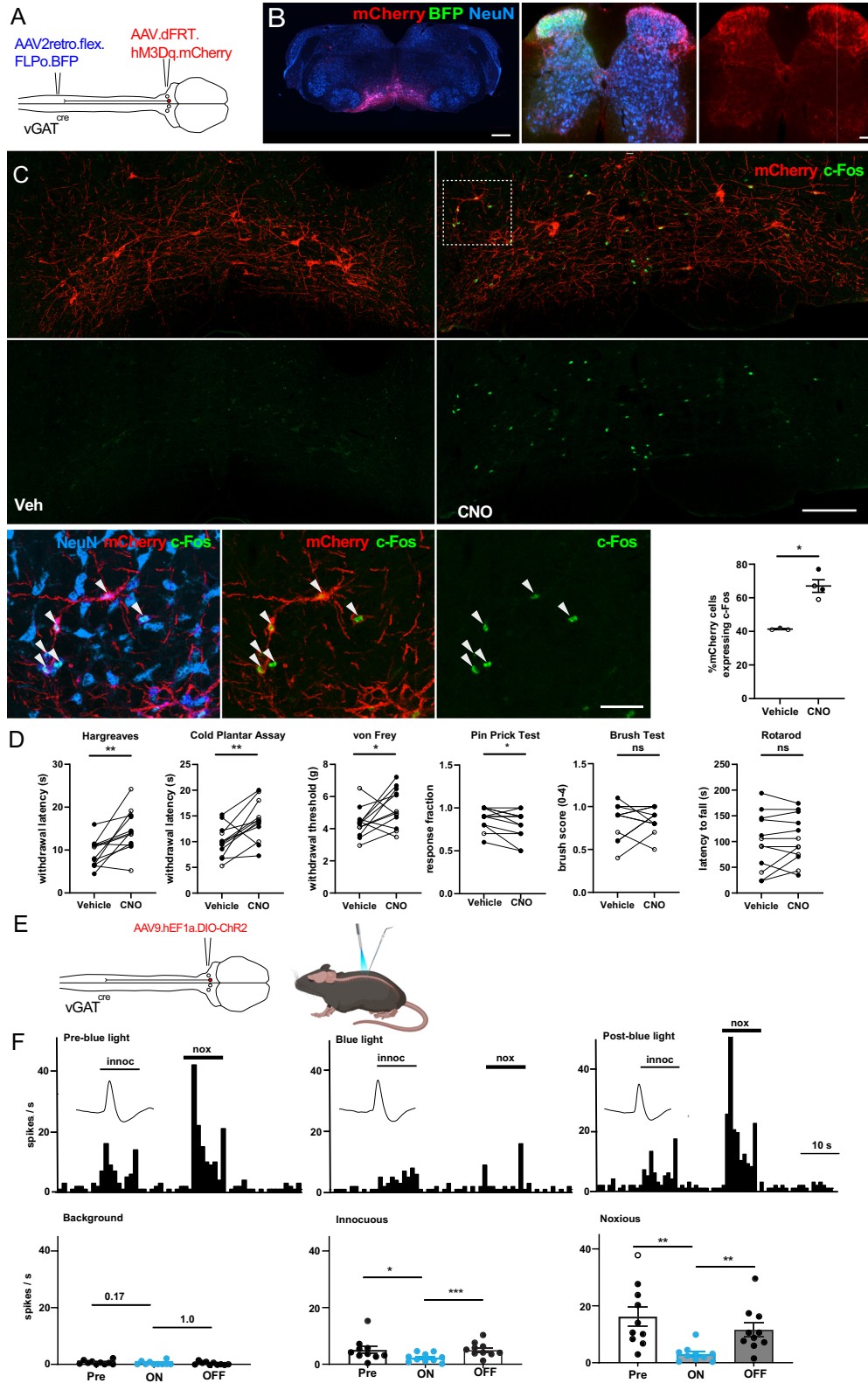

control, mCherry. We found significant increases in the withdrawal latencies and thresholds in the Hargreaves and cold plantar assays and electronic von Frey tests in hM3Dq-expressing animals relative to mCherry controls (Supplementary Fig. 2).

To verify that the antinociceptive effects described above depended on the activation of the descending spinal (rather than ascending brain) projections of vGAT RVM[SC] neurons, we followed two strategies. We first used local intrathecal injections of CNO (1 μg in 10 μl) or focal spinal optogenetic stimulation (0.2 ± 0.06 mW) (irradiance = 1.72 mW/mm² applied at 20 Hz for 1 s, every 10 s) of the lower lumbar spinal cord. Local CNO application increased withdrawal latencies from 9.55 ± 1.64 s to 19.11 ± 0.38 s ($p = 0.002$, paired $t$ test, $n = 5$), and 2.12 ± 0.05 s to 7.95 ± 1.22 s ($p = 0.008$, paired t test, $n = 5$), for heat and cold, respectively (Supplementary Fig. 3A–C). Focal

**Fig. 3 | Activation of descending vGAT RVM^SC neurons inhibits nociceptive sensitivity and reduces stimulus-evoked activity of dorsal horn WDR neurons. A** Injection scheme for introducing hM3Dq to vGAT RVM^SC neurons. **B** Micrographs showing injection sites. Scale bars, 200 μm (RVM), and 50 μm (spinal cord). **C** Top and middle: c-Fos immunoreactivity in hM3Dq-mCherry expressing RVM neurons (scale bar, 100 μm). Bottom: higher magnification of the boxed area. Arrowheads, mCherry⁺ neuron expressing c-Fos (scale bar, 20 μm). Bottom right, quantification of mCherry⁺ neurons upregulating c-Fos after CNO injection (unpaired two-sided t test, $p = 0.0022$, vehicle ($n = 3$ mice) vs CNO ($n = 4$ mice). Images in B and C are representative of $n = 12$ animals. **D** Behavioral effects of vGAT RVM^SC neuron activation. Two-sided paired t-tests. Hargreaves: $p = 0.0030$; cold plantar assay: $p = 0.0059$; von Frey test: $p = 0.042$, pin prick: $p = 0.032$; $n = 12$ mice). Brush responses (two-sided paired Wilcoxon test Brush vehicle vs CNO: $p = 0.78$; $n = 12$ mice). Rotarod test (paired t-test, $p = 0.39$, $n = 12$ mice per test). **E** Injection scheme

for introducing ChR2 to inhibitory RVM neurons (left) and optogenetic stimulation (right). **F** Top: peristimulus time histograms of action potentials (spikes/s) evoked by innocuous (innoc) and noxious (nox) mechanical stimulation of one hindpaw in a WDR neuron before (pre blue-light), during (blue-light on, 2 min) and after (blue-light off, 2 min) stimulation. Insets show individual action potential traces. Bottom: summary of effects. Optogenetic stimulation had no effects on background activity, but significantly inhibited neuronal activity evoked by innocuous (one-way repeated measures ANOVA followed by Bonferroni correction $F_{(1.2,11.0)} = 10.38$, $p = 0.0054$) and noxious ($F_{(1.3,11.3)} = 16.9$, $p = 0.0018$, $n = 10$) mechanical stimulation. After 2 min of blue light off, both innocuous ($p = 0.0007$) and noxious ($p = 0.0094$) mechanical stimulation evoked neuronal activity recovered. Bar histograms show mean ± SEM. *$p < 0.05$, **$p < 0.01$, ***$p < 0.001$. Open and closed symbols indicate female and male mice, respectively. Source data are provided as a Source Data file.

optogenetic stimulation yielded similar results. Withdrawal latencies increased from $8.50 \pm 0.92$ s to $13.43 \pm 1.11$ s ($p = 0.019$, paired t test, $n = 6$), and $6.72 \pm 0.40$ s to $7.84 \pm 0.30$ s ($p = 0.098$, paired $t$ test, $n = 6$), for heat and cold, respectively (Supplementary Fig. 3D–E). Both experiments demonstrate that selective stimulation of the spinal terminals of descending vGAT RVM neurons alone is sufficient to induce analgesia. Secondly, we combined in vivo electrophysiological recordings of superficial dorsal horn with optogenetic stimulation of the spinal terminals of RVM^SC neurons (Fig. 3E, F). We expressed ChR2 from a viral transgene (AAV9.hEF1a.Dio.ChR2) in inhibitory (vGAT^cre) neurons of the RVM and performed extracellular recordings from wide dynamic range (WDR) neurons of the superficial lumbar dorsal horn. Action potential firing was reliably evoked through noxious or innocuous stimulation of the respective cutaneous receptive fields. Blue light stimulation of the lumbar dorsal spinal cord reduced action potential firing evoked by noxious stimulation by $79.3 \pm 4.7\%$ ($n = 10$, $p = 0.0054$, one-way repeated measures ANOVA). Action potential firing in response to innocuous stimulation was also reduced albeit to a smaller extent ($45.3 \pm 5.4\%$, $n = 10$, $p = 0.043$). In control experiments, when mCherry (rAAV9.EF1a.Dio.mcherry) was expressed instead of ChR2, blue light stimulation did not change stimulation-induced action potential firing (Supplementary Fig. 3G, H). Taken together, these data indicate that activation of vGAT RVM^SC neurons inhibits sensitivity to noxious thermal and mechanical stimuli through an action that involves descending inhibitory control of spinal nociceptive processing.

## vGAT RVM^SC projections form functional synapses throughout the spinal cord and can mediate widespread thermal antinociception

The presence of wide-ranging axon collaterals in multiple spinal segments raises the possibility that these neurons control nociception, not only bilaterally, but also along the rostrocaudal axis. To address this question, we again used an optogenetic approach and labeled descending vGAT RVM^SC neurons with ChR2 from lumbar spinal cord injection sites and prepared slices from both the lumbar and cervical spinal cord of these mice. Subsequently, we tested whether light-evoked inhibitory postsynaptic currents (IPSCs) could be recorded both from lumbar and cervical superficial dorsal horn neurons (Fig. 4A). In agreement with the previous tracing experiments (Fig. 2), light-evoked IPSCs were present in both lumbar and cervical neurons. Light-evoked IPSCs from both locations were consistently blocked by co-application of bicuculline (20 μM) and strychnine (0.5 μM) (Fig. 4B, C). Separate application of either bicuculline or strychnine significantly reduced the amplitude of the light-evoked IPSCs (Supplementary Fig. 4), confirming the contribution of both transmitters.

We next asked whether this wide-ranging innervation would manifest in similarly wide-spread analgesia. We used the same expression strategy as above but introduced hM3Dq for chemogenetic

activation (Fig. 4D–G). First experiments addressing possible contralateral effects revealed significant changes in heat and cold sensitivity in the contralateral paw, but not in the von Frey filament assay in the contralateral paw (Supplementary Fig. 5). In subsequent experiments, we tested effects on heat and cold sensitivity in all four paws. In the Hargreaves test, we found significant CNO-induced latency changes for all four paws (Fig. 4E). In the cold plantar assay, heterotopic effects in the contralateral hindpaw and ipsilateral forepaw were significant but less pronounced (Fig. 4F). Finally, we tested whether the activation of the vGAT RVM^SC neurons was also able to inhibit sensitivity of the facial areas to nociceptive stimuli. We applied droplets of the TRPV1 agonist capsaicin, which elicits burning pain sensations through activation of TRPV1 positive sensory neurons[28], to the eyes of mice. Nociceptive activation was quantified as the number of forepaw wipes directed towards the capsaicin exposed eye. Chemogenetic activation of descending vGAT RVM^SC neurons traced from the lumbar spinal cord reduced the number of forepaw wipes when compared to vehicle-injected controls (Fig. 4G). Taken together, these data indicate that activation of descending vGAT RVM^SC neurons inhibits nociceptive sensitivity in a near body-wide manner.

## Silencing of vGAT RVM^SC neurons evoked mechanical hypersensitivity, tactile allodynia, and spontaneous aversive behavior

Chemogenetic activation experiments demonstrated that excitation of descending vGAT RVM^SC neurons reduces sensitivity to nociceptive stimuli, but whether these neurons also regulate baseline sensitivity remained to be explored. To this end, we performed loss-of-function experiments and used our intersectional approach to express tetanus toxin light chain (TetLC) in vGAT RVM^SC neurons (Fig. 5A–C). Within 7 days of virus injection, TetLC-expressing animals developed strong hypersensitivity to both punctate and dynamic mechanical stimulation (Fig. 5D). Strikingly, these animals exhibited spontaneous aversive behaviors, such as flinching of the ipsilateral hindpaw, and a reluctance to use this paw for locomotion (Supplementary Video 2). Gross sensorimotor coordination assessed in the rotarod test remained unaltered. In contrast to the chemogenetic activation experiments, silencing the vGAT RVM^SC neurons did not significantly alter heat or cold sensitivity, suggesting that these neurons are not involved in the control of heat and cold sensitivity in healthy mice. Although the observed changes largely affected the injected side, we also observed a slight increase in sensitivity to dynamic mechanical stimuli on the contralateral paw, consistent with our anatomical labeling experiment (compare Fig. 3).

These data suggest that vGAT RVM^SC neurons include populations whose activation can inhibit nociceptive sensitivity, and other populations that are tonically active under resting conditions to control mechanical sensitivity. Accordingly, we hypothesized that there are neurons within the vGAT RVM^SC population with lower basal activity that can be recruited to reduce nociceptive sensitivity, and others that

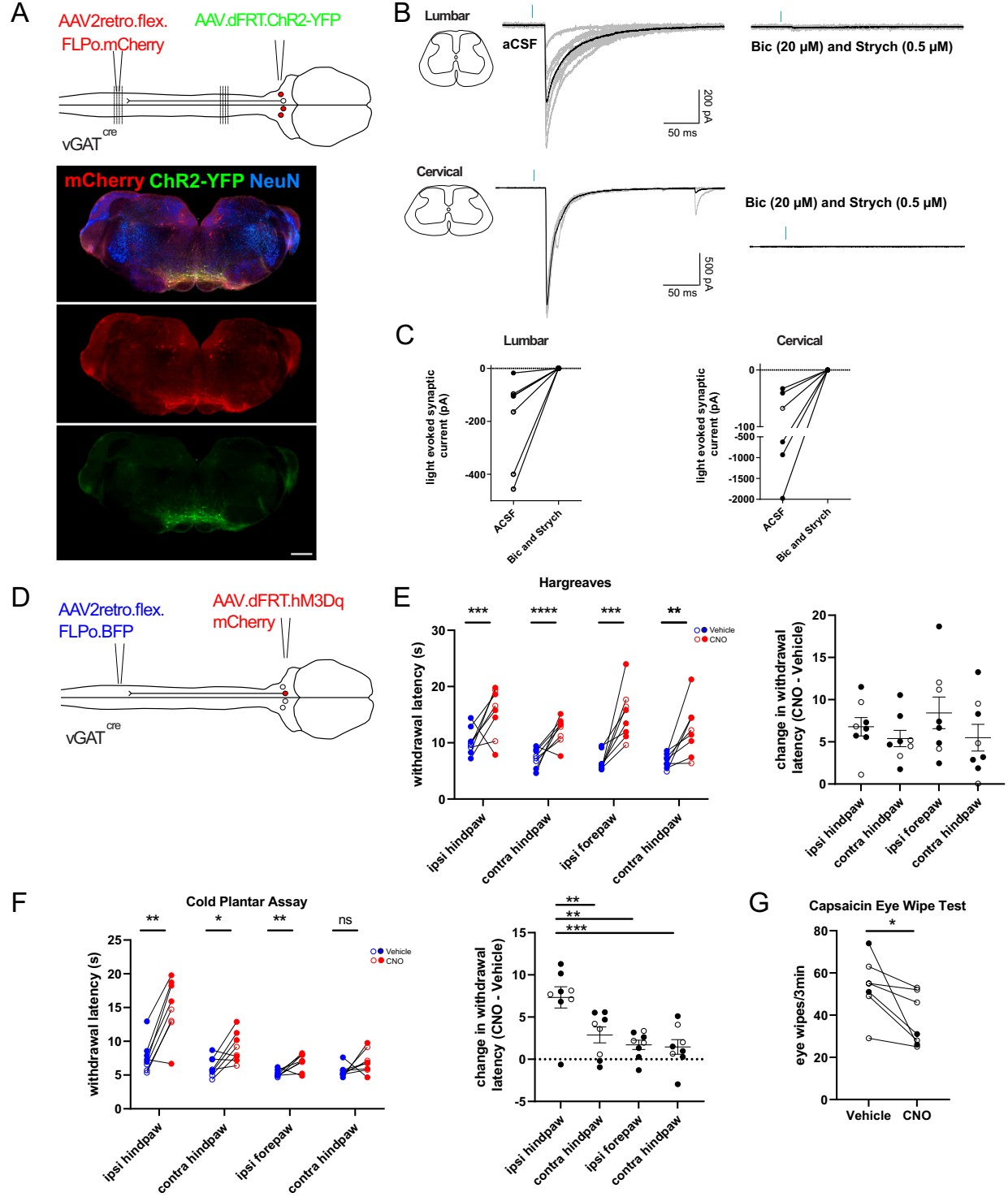

**Figure (A–G)**

have tonic activity and prevent mechanical hypersensitivity/allodynia. To address this, we performed targeted whole-cell recordings from retrogradely labeled vGAT RVM$^{SC}$ neurons (Fig. 5E–H). Recorded neurons could be broadly divided into three categories based on their spontaneous action potential firing patterns. These included spontaneously active neurons with regular or irregular firing, and inactive neurons (referred to as regular, irregular and silent respectively) (Fig. 5G), consistent with the findings of others[29]. Regular and irregular firing neurons could be distinguished based on the coefficient of variance of their interspike intervals (CV$_{ISI}$) (Fig. 5H)[30]. Regular firing neurons exhibited firing frequencies that varied between 3.2 and

14.7 Hz and were consistently higher than those of irregular firing neurons (0.13–0.77 Hz).

## Recruitment of vGAT RVM$^{SC}$ neurons can reverse thermal and mechanical hypersensitivity

Silencing vGAT RVM$^{SC}$ neurons produced mechanical hypersensitivity that was reminiscent of that seen in pathological pain states. We therefore hypothesized that increased activation of these neurons could reverse hypersensitivity in inflammatory and neuropathic pain models. To test this possibility, we induced peripheral inflammation by injecting Complete Freund's adjuvant (CFA) into the left hindpaw and

**Fig. 4 | vGAT RVM^SC neurons form functional synapses throughout the spinal cord and globally reduce thermal sensitivity. A** Top, intersectional strategy for expressing ChR2 in vGAT RVM^SC neurons from the lumbar spinal cord. Micrographs show an RVM injection site from intersectional labeling of vGAT RVM^SC neurons (scale bar, 200 μm). Similar results were obtained from $n = 4$ mice. **B** Synaptic currents evoked by blue light illumination recorded from neurons in lumbar and cervical spinal cord slices. **C** Group data (cervical, $n = 6$ neurons; lumbar, $n = 7$ neurons). **D** Intersectional strategy for expressing hM3Dq in vGAT RVM^SC neurons that project to the left lumbar spinal cord. **E** Withdrawal latencies of paws to heat stimuli following CNO or vehicle injection (ipsilateral hindpaw: $p = 0.00033$, contralateral hindpaw: $p = 0.000072$, ipsilateral forepaw: $p = 0.00049$, contralateral forepaw $p = 0.0064$, $n = 4$ mice, multiple two-sided unpaired $t$-tests with Holm-Sidak correction, adjusted $p$ values). This increase in withdrawal latencies is

comparable between the ipsilateral hindpaw and the other paws (repeated measures one way ANOVA $F_{(3,21)} = 1.208$, $p = 0.33$). All data are displayed with mean ± SEM indicated. **F** Cold plantar assay. Left, vehicle versus CNO. Ipsilateral hindpaw $p = 0.0032$, contralateral hindpaw $p = 0.016$, ipsilateral forepaw $p = 0.0097$, contralateral forepaw $p = 0.051$, adjusted $p$ values) (repeated two-sided $t$-tests with adjusted $p$ values, Holm-Sidak post hoc correction). Right, ipsilateral hindpaw versus other paws. Repeated measures one way ANOVA $F_{(3,21)} = 8.46$, $p = 0.0007$, $n = 8$ mice. All data are displayed with mean ± SEM indicated. **G** Number of forepaw wipes directed towards the affected eye (two-sided paired $t$ test, $p = 0.032$, $n = 7$ mice). *, $p < 0.05$; **, $p < 0.01$; ***, $p < 0.001$; ****, $p < 0.0001$; ns, not significant. Open and closed symbols indicate female and male mice, respectively. Source data are provided as a Source Data file.

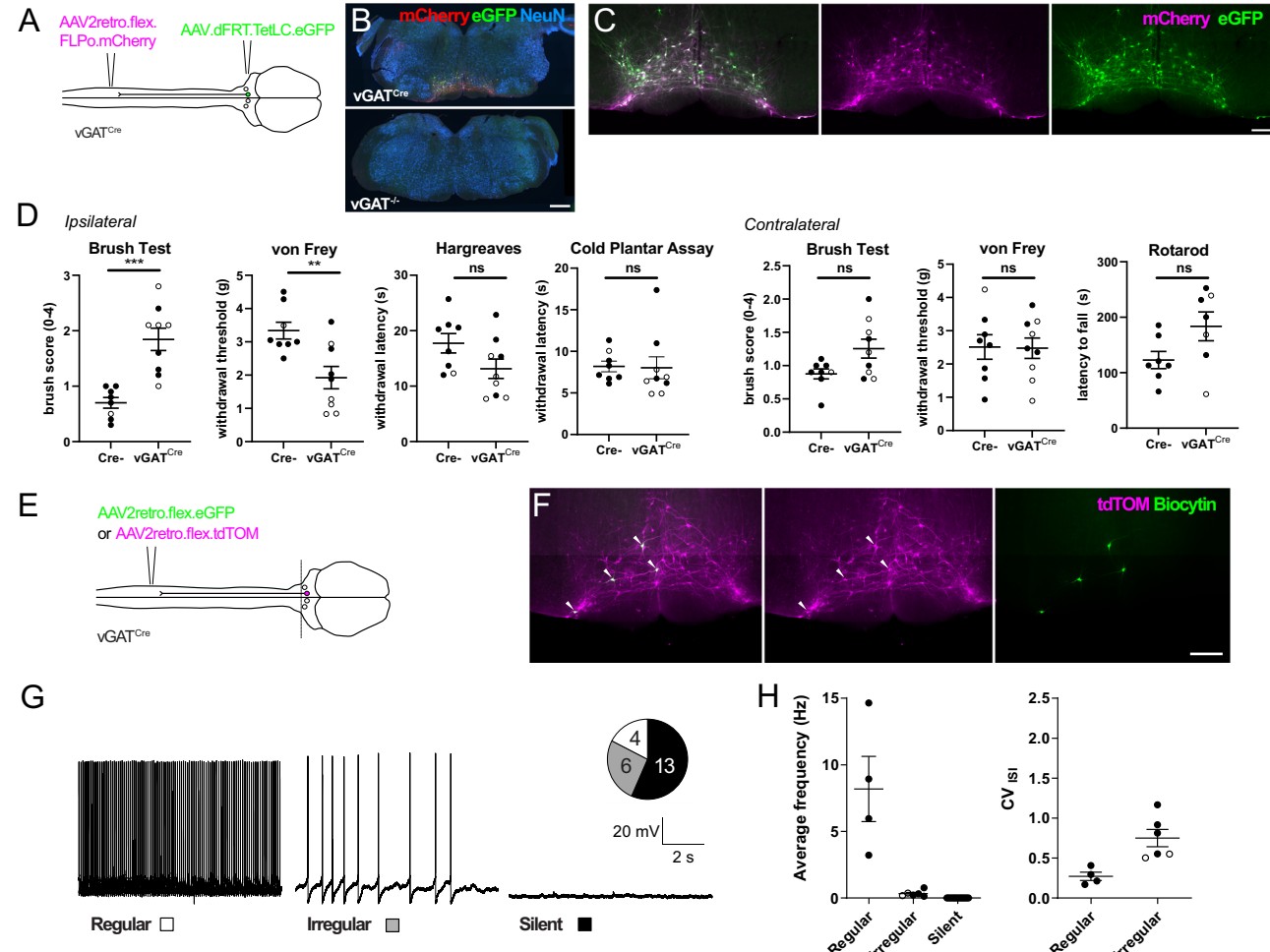

**Fig. 5 | Tetanus toxin-mediated silencing of vGAT RVM^SC neurons produces mechanical allodynia. A** Injection scheme for silencing vGAT RVM^SC neurons with TetLC. **B** Brainstem sections with example RVM injection sites from vGAT^Cre mice and cre⁻ littermates. Scale bar, 500 μm. **C** RVM at higher magnification. Scale bar, 100 μm. Similar results were obtained in $n = 9$ vGAT^Cre mice and $n = 8$ wild-type littermates. **D** Behavioral testing (Cre⁻ ($n = 9$) versus vGAT^cre mice ($n = 8$). Ipsilateral paw: von Frey $p = 0.0002$ (two-tailed unpaired $t$-test); brush $p = 0.0047$ (unpaired Mann-Whitney test); Hargreaves $p = 0.084$ (two-sided unpaired $t$-test); cold plantar $p = 0.32$ (two-sided unpaired $t$-test). Contralateral paw: brush $p = 0.12$ (two-sided unpaired Mann Whitney test); von Frey $p = 0.94$ (two-sided unpaired $t$-test:). Rotarod assay $p = 0.068$ (unpaired $t$-test). **, $p < 0.01$; ***, $p < 0.001$; ns, not significant. All data are displayed together with mean ± SEM. **E** Injection scheme for labeling vGAT RVM^SC neurons for targeted whole-cell recordings. **F** Hindbrain slice

containing tdTomato⁺ cells labeled with biocytin during recordings (scale bar, 100 μm). Hindbrain section is representative of sections from $n = 4$ mice.
**G** Representative traces from vGAT RVM^SC neurons recorded in current clamp mode. Inset illustrates the frequency at which regular and irregular firing and silent neurons were observed. **H** Average frequency of spontaneous action potential firing for each neuron type, and the coefficient of variance for the interspike intervals (CV_{ISI}) of irregular and regular firing vGAT RVM^SC neurons ($n = 23$ cells from 4 mice; regular = 4 neurons from 3 mice, irregular = 6 neurons from 4 mice, silent = 13 neurons from 4 mice). Regular neurons have a consistently lower CV_{ISI} than irregular neurons. Open and closed symbols indicate female and male mice, respectively. All data are displayed together with mean ± SEM. Source data are provided as a Source Data file.

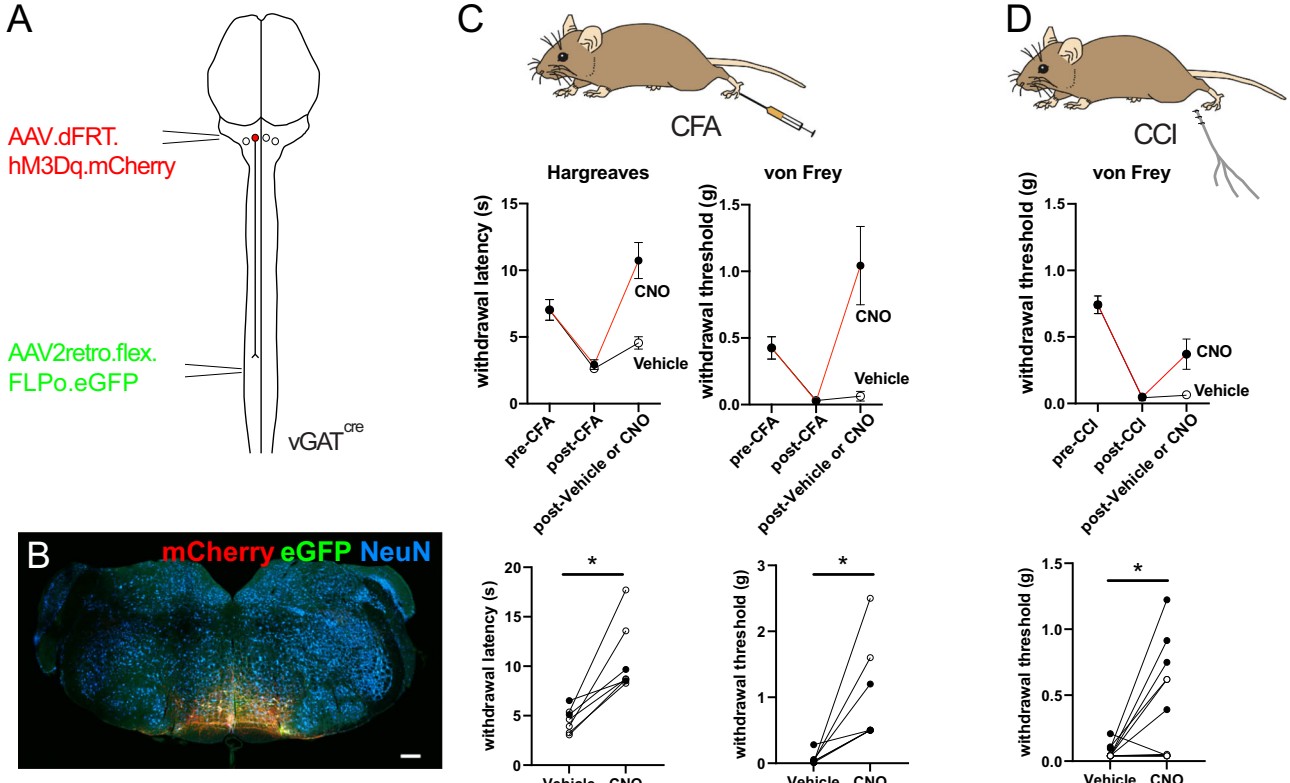

**Fig. 6 | Chemogenetic activation of descending vGAT RVM$^{SC}$ neurons reverses pathological hypersensitivity. A** Injection scheme to express hM3Dq in descending vGAT RVM$^{SC}$ neurons. **B** Representative RVM injection site from vGAT RVM$^{SC}$ chemogenetic activation experiments (scale bar, 200 μm). Hindbrain injection site is representative of $n = 20$ animals used in behavioral analyses. **C** CFA injection induced mechanical and thermal hypersensitivity that was reversed by activation of vGAT RVM$^{SC}$ neurons with CNO (2-way repeated measures ANOVA, significant interaction of treatment x time Hargreaves ($F_{(2,24)} = 13.41$, $p = 0.0001$) and von Frey ($F_{(2,24)} = 10.6$, $p = 0.0005$, $n = 7$ mice). Data are summarized as mean ± SEM. Below, comparison of post-vehicle or CNO injection in hypersensitive animals, indicating the change in withdrawal thresholds and latencies for each animal (two-sided Wilcoxon matched-pairs signed rank test: Hargreaves $p = 0.0156$, von Frey $p = 0.016$). **D** Mechanical allodynia was induced by CCI and this was reversed by CNO injection (mixed-effects analysis, significant interaction of treatment x time $F_{(2,48)} = 4.40$, $p = 0.018$). Below, comparison of von Frey thresholds in neuropathic animals that received CNO and vehicle injection (two-sided Wilcoxon matched-pairs signed rank test $p = 0.020$, $n = 13$ mice. Data are summarized as mean ± SEM. *, $p < 0.05$; **, $p < 0.01$; ***, $p < 0.001$). **C, D** (bottom panels), open and closed symbols indicate female and male mice, respectively. Source data are provided as a Source Data file.

repeated the chemogenetic activation experiments (Fig. 6A, B). Intraplantar injection of CFA increased mechanical and heat sensitivity, which was reversed by activation of vGAT RVM$^{SC}$ neurons (Fig. 6C). Strikingly, activation of vGAT RVM$^{SC}$ neurons was also sufficient to reverse CFA inflammation-induced hypersensitivity in the contralateral hindpaw (Supplementary Fig. 6). To determine whether a similar reversal of allodynia could also be achieved in pathological pain states, we performed chemogenetic activation experiments in mice with a chronic constriction injury (CCI) of the sciatic nerve[31]. Activation of vGAT RVM$^{SC}$ neurons reversed mechanical allodynia measured on day 7 after nerve injury (Fig. 6D). Therefore, vGAT RVM$^{SC}$ neurons are not only required for normal mechanical sensitivity but are also able to reverse mechanical allodynia in hyperalgesic disease states.

## vGAT RVM$^{SC}$ neurons are engaged during ongoing pain and provide feed-forward inhibition of nociception

Our data suggest that vGAT RVM$^{SC}$ neurons can broadly inhibit pain sensitivity throughout the body. We asked under what circumstances these neurons are engaged, and hypothesized that they might contribute to the so-called pain-inhibits-pain phenomenon. It is well-known that pain sensitivity can be reduced by other painful stimuli applied to a remote, e.g., contralateral site of the body. The involvement of descending inhibitory projections in this "pain-inhibits-pain" is well established[7,32,33]. Neurons involved in this phenomenon are

predicted to have at least three properties. They should be activated by painful stimuli, provide wide-spread analgesia, and be capable of reducing pain when active[9]. Since vGAT RVM$^{SC}$ neurons contain wide-ranging axonal projections and their activation reduces nociception, we next asked whether they are activated by pain. To address this, we labeled vGAT RVM$^{SC}$ neurons from the lumbar spinal cord and injected capsaicin into the right forepaw (20 μl capsaicin, 1 mg/ml) (Fig. 7A). We found that $17.8 \pm 0.8\%$ of labeled vGAT RVM$^{SC}$ neurons upregulated the activity-dependent marker c-Fos in response to capsaicin injection (Fig. 7B, C). Interestingly, these activated neurons were largely restricted to the LPGi (Fig. 7D).

To directly test if the same neurons activated by a painful stimulus provide wide-spread analgesia, we employed the TRAP technique and used TRAP2 (Fos$^{2A-CreERT2}$) mice that express a tamoxifen dependent Cre recombinase under the transcriptional control of the activity-dependent marker gene c-Fos. We combined this approach with an intersectional viral strategy to restrict the expression of reporter or actuator transgenes to RVM$^{SC}$ neurons activated by noxious stimuli (Fig. 8A). We first characterized the neurochemistry of noxious stimulus activated RVM$^{SC}$ TRAP2 neurons (TRAP2$^{caps}$ RVM$^{SC}$) using mFISH. To enable visual identification of TRAP2$^{caps}$ RVM$^{SC}$ neurons, we injected the lumbar spinal cord of TRAP2;Ai65 (cre and flp dependent tdTomato) double transgenic mice with AAV2retro.flex.FLPo. Two weeks later, capsaicin was injected into the right forepaw, followed by a

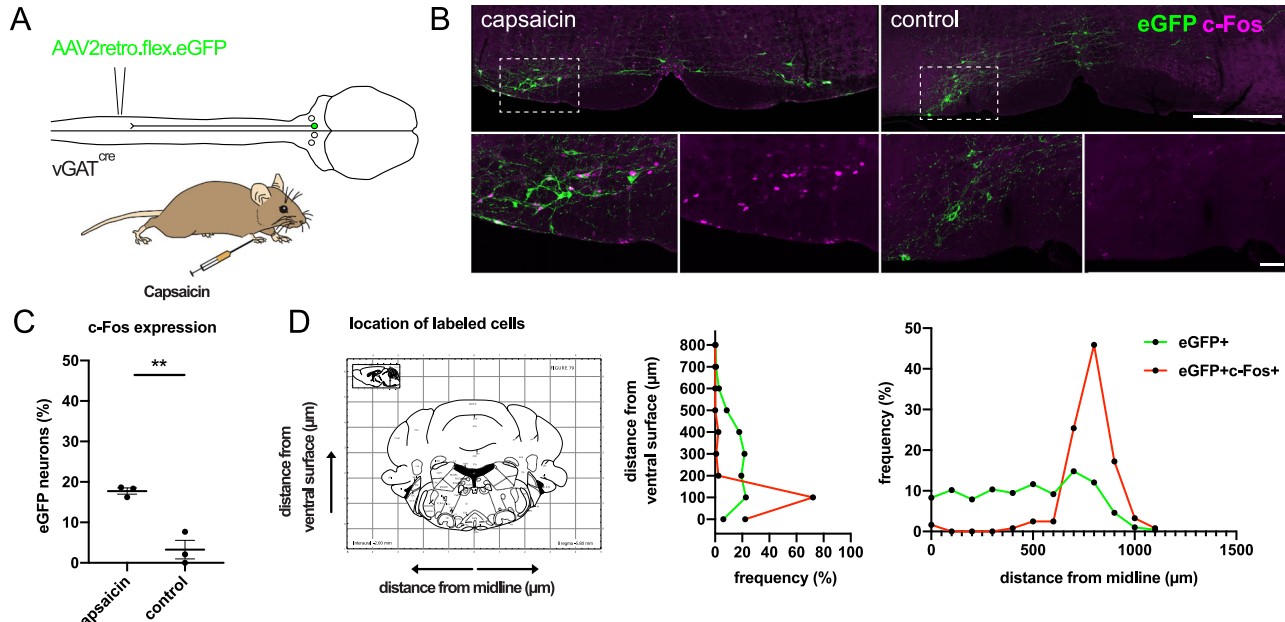

**Fig. 7 | Descending vGAT RVM$^{sc}$ neurons in the LPGi are activated by noxious forepaw stimulation. A** Injection scheme for retrograde labeling of descending inhibitory projection neurons from the lumbar spinal cord and noxious stimulus delivery to the right forepaw. **B** Example of hindbrain tissues taken from capsaicin stimulated and control animals (scale bar, 200 μm, insets scale bar, 20 μm, images are a maximum intensity projection of 2 optical sections at 5 μm z-spacing). **C** A subset of vGAT RVM$^{sc}$ inhibitory projections upregulates c-Fos following capsaicin injection into the right forepaw. Capsaicin, 17.8%; control, 3.3%; two-sided unpaired *t*-test 0.0039, capsaicin *n* = 3 mice (177–314 cells), control *n* = 3 mice (142–247 cells). Data are displayed together with mean ± SEM. **D** Left: illustration for the measures used to determine the position of labeled neurons in the RVM relative to anatomical landmarks, taken from the mouse brain atlas in stereotaxic coordinates. Right: location of all labeled vGAT RVM$^{sc}$ neurons (green), and those that upregulate c-Fos (red) after forepaw capsaicin injection. Note that the c-Fos-labeled cells are located in the LPGi, closer to the ventral surface and further from the midline than the general population of descending vGAT$^{cre}$ neurons. **, *p* < 0.01. Open and closed symbols indicate female and male mice, respectively. Source data are provided as a Source Data file.

4-OHT injection 1 h later. Another 2 weeks later, hindbrains from these animals were prepared for mFISH experiments (Fig. 8B). Most of the capsaicin-activated neurons contained vGAT (74.9%), a much higher proportion than non-specific labeling with AAV2retro.eGFP (Fig. 8C, see also Fig. 1A), indicating that most TRAP2$^{caps}$ RVM$^{SC}$ neurons were inhibitory. In agreement with the previous c-Fos labeling experiments (Fig. 7D), the majority of neurons were found in the LPGi (Fig. 8E). Also, in agreement with the previous labeling experiments, axon terminals of these neurons were most abundant in the superficial dorsal horn (Fig. 8F) and virtually all mCherry$^+$ axonal boutons contained detectable levels of vGAT (Fig. 8G). When capsaicin was injected into the forepaw again, many (58.2 ± 2.5%) of the labeled neurons in the RVM upregulated c-Fos, indicating they were consistently activated by noxious stimuli (Fig. 8H, I). We then tested whether chemogenetic reactivation of the TRAP2$^{caps}$ RVM$^{SC}$ neurons would reduce nociception in the contralateral hindpaw, i.e., to a site distant from the initial stimulation (ipsilateral forepaw) (Fig. 8J). Sensitivity to noxious heat and cold stimuli was significantly reduced, while responses to mechanical stimuli were not altered. These findings demonstrate that vGAT RVM$^{SC}$ neurons in the LPGi are activated by noxious stimuli, and in turn reduce nociception in a wide-spread manner.

## Discussion

Much research on descending pain modulation has focused on monoaminergic neurotransmission and on neuromodulators such as opioid peptides and endocannabinoids[7,34–37]. These systems are readily accessible with pharmacological approaches while this is not the case for pathways that use fast neurotransmitters such as GABA and glycine. The latter transmitter systems are almost ubiquitously present in the CNS, greatly hampering pharmacological approaches. Intersectional

genetics employing recombinase dependent transgenes locally delivered through viral vectors allow their selective targeting with great precision[8,10,14,20,38].

In the present study, we specifically addressed inhibitory projections reaching the spinal cord from the RVM and employed intersectional approaches that made use of vGAT$^{cre}$ mice injected at the spinal level with AAV2retro carrying a cre-dependent Flpo expression cassette. Flpo dependent reporter and actuator constructs were then expressed from a second AAV injected into the RVM. This approach permitted the targeted recording, and specific tracing, chemogenetic and optogenetic activation, and silencing of inhibitory projections from the RVM to the spinal dorsal horn. Using these tools, our study identified a population of descending inhibitory RVM neurons that have wide-ranging axonal projections terminating bilaterally throughout the rostrocaudal axis of the spinal cord. These neurons, or a subset of them located mainly in the LPGi, contribute the pain-inhibits-pain phenomenon both in naïve mice and in mice in hyperalgesic disease states.

vGAT RVM$^{SC}$ neurons not only project caudally to the spinal cord but also send ascending axon collaterals to supraspinal sites. Inhibitory effects of electrical or chemical RVM stimulation on tail-flick[39] or von Frey filament-induced[40] hind paw withdrawal responses, as well as on the suppression of noxious stimulus-induced spinal neuron firing[41,42], have previously been shown to require intact dorsolateral funiculi. Our behavioral experiments with local chemogenetic or optogenetic activation of vGAT RVM$^{SC}$ neurons and the dorsal horn WDR neuron recordings directly demonstrate that activation of the descending projection is sufficient to evoke behavioral analgesia and to reduce dorsal horn WDR neuron firing. The latter result also confirms that the vGAT RVM$^{SC}$ neuron stimulation does not only impact withdrawal

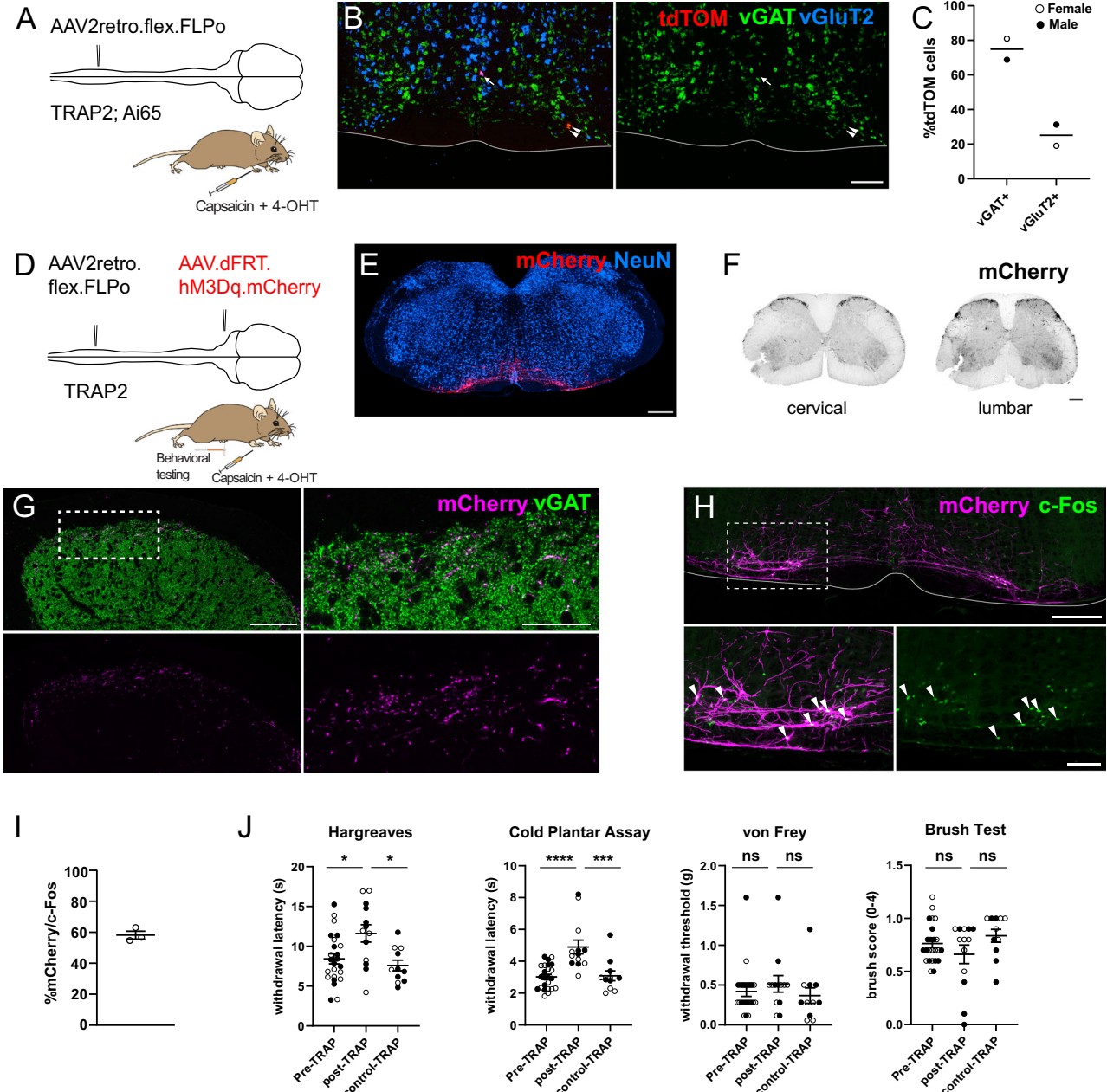

**Fig. 8 | Noxious stimulus-activated RVM^sc neurons are mainly inhibitory, project to the superficial dorsal horn, and inhibit noxious temperature sensitivity.**
**A** Injection scheme for labeling TRAP2^caps RVM^sc neurons activated by forepaw injection of capsaicin. **B** Micrographs of tdTomato-labeled neurons in the RVM of TRAP2;Ai65 mice containing vGAT or vGluT2 mRNA. Arrowheads indicate vGAT^+ cells and arrows indicate vGluT2^+ cells (scale bar, 200 μm). **C** Quantification of TRAP2^caps RVM^sc neurons containing vGAT or vGluT2 mRNA (74.9% and 25.1% respectively, n = 2 mice). **D** Injection scheme for labeling and reactivating TRAP2^caps RVM^sc neurons. **E** Representative hindbrain section from a TRAP2 mouse that received the injections depicted in (**D**) (scale bar, 500 μm). **F** Spinal cord sections taken from the animal shown in E (scale bar, 200 μm). **G** Representative micrographs of a lumbar spinal cord section containing mCherry-labeled axons from TRAP2^caps RVM^sc neurons containing vGAT. Scale bar, 100 μm; or 50 μm (inset).

Micrographs displayed in **E−G** are representative of n = 13 mice used in behavioral analyses (J) that received the injections depicted in **D** followed by forepaw capsaicin and 4-OHT. **H** Example hindbrain sections taken from a mouse that received the same injections depicted in D, and a second forepaw capsaicin injection Many mCherry-labeled cells upregulate c-Fos (arrowheads). Scale bar, 200 μm; inset scale bar, 100 μm. **I** Quantification of mCherry-labeled neurons that upregulate c-Fos after a second capsaicin injection (n = 3 mice). Data are displayed together with mean ± SEM. **J** Behavioral analyses. Chemogenetic re-activation of TRAP2^caps RVM^sc neurons increases hindpaw withdrawal latencies to heat and cold plantar stimulation (Hargreaves one-way ANOVA p = 0.005). All measurements were taken 1–3 h after CNO injection (n = 24 mice (pre-TRAP), n = 13 (post-TRAP), n = 11 (control-TRAP)). Data are displayed together with mean ± SEM. *, p < 0.05; ***, p < 0.001; **** p < 0.0001, ns, not significant. Source data are provided as a Source Data file.

responses but reduces nociceptive processing at the level of the dorsal horn.

Seminal work on the RVM has identified so called ON, OFF and NEUTRAL cells on the basis of their activity around the time of a nocifensive (heat-induced tail flick) reaction[43]. ON cells increase firing,

OFF cells decrease firing and NEUTRAL cells show no change in activity. It has been proposed that OFF cells must pause activity to allow the nociceptive response to occur[43], but see also refs. 44,45. Accordingly, activation of OFF cells is believed to be analgesic and, although so far no genetic markers have been identified, many OFF

cells appear to be GABAergic[46,47]. The results of our chemogenetic activation experiments are consistent with the idea that the vGAT RVM[SC] neurons studied here are mainly OFF cells and exert a net antinociceptive action. Furthermore, manipulations that decrease the activity of OFF cells have previously been shown to lead to a general enhancement of nocifensive responses[48], which mirrors the occurrence of allodynia induced by the silencing of vGAT RVM[SC] neurons in our study.

Consistent with the presence of analgesic OFF and pronociceptive ON cells, behavioral and electrophysiological studies have found that electrical or chemical stimulation of the RVM can have inhibitory, faciliatory or biphasic effects[39,49]. Chemogenetic activation of the vGAT RVM[SC] neurons studied here was exclusively antinociceptive and it is likely that the faciliatory and biphasic actions originate from different RVM neuron populations, e.g., ON and OFF cells[50]. This is also supported by studies suggesting that activation of neurons of the LPGi inhibits nociception whereas activation of those in the nucleus raphe magnus is facilitatory[20]. Other studies have shown that inhibition of nociception requires intact dorsolateral funiculi[40], whereas facilitatory effects require the ventral/ventrolateral funiculi[39,40]. Consistent with these findings, the descending axons of the inhibitory RVM projections that caused antinociception in the present study are found in the dorsolateral funiculi.

Previous studies that employed electrical or chemical stimulation of the RVM found inhibitory, faciliatory and biphasic effects on noxious stimulus-evoked nocifensive responses[39,40], and firing of spinal neurons[41,42]. Two studies reported that weaker stimulation intensities favored facilitation and stronger ones inhibition[39,40]. When we employed chemogenetics or optogenetics to selectively target descending inhibitory RVM neurons, inhibitory effects were observed in the great majority of experiments, especially in those with thermal stimulation (Figs. 3 and 4; and Supplementary Figs. 5 and 6). A slightly higher heterogeneity was observed for mechanical stimuli (Fig. 3 and Supplementary Fig 5). We assume that the difference between previous studies and our study originates from differences in the targeted neuron populations.

Chemogenetic activation and tetanus toxin-mediated silencing of vGAT RVM[SC] neurons did not exactly mirror each other. Activation produced wide-spread analgesia, which was most prominent for thermal stimuli, while silencing produced a more localized and preferentially mechanical sensitization. It is possible that different subsets of vGAT RVM[SC] neurons preferentially innervate different dorsal horn areas, with those descending neurons that modulate primarily thermal nociception terminating in the superficial laminae whereas those that modulate primarily mechanical sensitivity would terminate more deeply. Our morphological tracing experiments have indeed shown that, although innervation by vGAT RVM[SC] neurons is most dense in the superficial layers, it is also present in the mechanosensitive deep dorsal horn.

Our slice recordings support that vGAT RVM[SC] neurons comprise potentially several subsets. In slice recordings many neurons were found tonically active while others were less active, consistent with previous in vivo work[47,51]. Different subsets of vGAT RVM neurons have also been identified by other groups. Zhang et al.[34] investigated an RVM population that was both GABAergic and enkephalinergic and found that their chemogenetic activation reduced both noxious mechanical and heat sensitivity, while silencing the same neurons with tetanus toxin enhanced nociception. These results and similar results of others[52] are in good agreement with our findings. François et al.[14] investigated the role of descending GABAergic RVM neurons as regulators of pro-enkephalinergic (pENK) dorsal horn neurons. They found that chemogenetic and optogenetic inhibition of vGAT RVM[SC] neurons reduced, rather than enhanced, mechanical sensitivity, but had no effect on heat thresholds. Differences between our results and those of François et al. are likely explained by the targeting of different

parts of the RVM due to differences in injections. François et al. used a single mid-line virus injection to target the RVM, whereas in our experiments, we used bilateral injections (Supplementary Fig. 1). In our experience, single mid-line injections largely miss the LPGi. This further supports our notion that different populations of vGAT RVM[SC] neurons can tune nociception in different directions.

As demonstrated in the chemogenetic reactivation experiments, a subset of vGAT RVM[SC] neurons are activated by nociceptive stimuli and, in turn, induce wide-spread analgesia upon reactivation, consistent with a role in pain-inhibits-pain phenomena. Related phenomena, which also require the engagement of descending systems[42,53], have been extensively studied both in rodents and humans. DNIC describes a phenomenon studied in vivo usually with single unit recordings from WDR dorsal horn neurons[33]. It is defined by the inhibition of these neurons by noxious stimuli delivered to body sites far away from the receptive field of the recorded neuron. A related phenomenon in humans is called CPM. It is measured as a reduction in reported pain intensity of a test stimulus through a conditioning stimulus again applied to a distant site[16]. The pain-inhibits-pain system we describe here is also inhibitory and provides wide-spread ("diffuse") inhibitory control over nocifensive behavior and reduces the response of dorsal horn WDR neurons to nociceptive input. It is therefore conceivable that it underlies at least some forms of DNIC and CPM. In fact, previous work also found evidence for an involvement of RVM OFF cells in DNIC[54].

Clinical studies suggest that the efficacy of CPM may be reduced in chronic pain states[16]. In our study, we found that antinociceptive actions of vGAT RVM[SC] neuron activation were fully retained in mice with inflamed paws, while hyperalgesia in the mice with a CCI surgery was only partially reversed. This may indicate that the efficacy of descending pain modulation is reduced with chronic neuropathic pain conditions or, possibly, restricted to the inflammatory component present in the CCI model.

While many forms of chronic pain are accompanied by regional hyperalgesia and allodynia, often generated through local peripheral or segmental spinal changes, other pain syndromes, such as in fibromyalgia, lead to more widespread pain affecting multiple body regions[55–58]. Widespread pain symptoms can also be observed in patients after spinal cord injury[59,60] and in patients suffering from multiple sclerosis[61–63]. Pain in these patients is mainly of central neuropathic origin[64]. Many spinal cord injury patients develop mechanical and thermal hypersensitivity below the site of injury[65,66], and multiple sclerosis patients exhibit wide-spread spontaneous pain and mechanical and thermal hypersensitivity[66–69]. These patients thus display symptoms similar to those observed in our experiments after silencing the vGAT RVM[SC] pathway in mice. It is hence tempting to speculate that the widespread pain in these patients may originate from a dysfunction of the vGAT RVM[SC] pathway caused by injury or demyelination.

In summary, our study has shown that vGAT RVM[SC] neurons are critical elements of a circuit controlling physiological pain sensitivity and are capable of powerful suppression of nociception under physiological and pathological conditions. Subsets of these neurons are central elements of a circuit for pain-mediated inhibition of pain and may thus be involved in DNIC and CPM. Since silencing these neurons leads to mechanical allodynia, this pathway may also contribute to widespread pain in patients with traumatic or inflammatory spinal cord disorders.

## Methods
### Animals
Mice aged between 6 and 12 weeks were used for behavioral and anatomical labeling experiments. For electrophysiology experiments, animals aged between 3 and 4 weeks were injected and were prepared for slice recordings >10 days later. Experiments were performed both in female and male mice. Different (opened and filled) symbols were

used to identify data points obtained from female and male mice. Experiments were not powered to identify sex-specific differences. Additional experiments would be required to test whether such differences exist.

Permission to perform animal experiments was obtained either from the Veterinäramt des Kantons Zürich (154/2018 and 097/2021) or were approved by the National Institute of Dental and Craniofacial Research (NIDCR) ACUC and followed National Institutes of Health (NIH) guidelines. All transgenic mouse lines used in this study are listed in the Supplementary Table 5. Animals were housed in a temperature- (21–24°C) and humidity- (40–60%) controlled facility on a 12 h:12 h inverted light cycle, with ad libitum access to food and water.

### Experimental designs

Whenever possible, animals were randomly attributed to treatment group. When different genotypes were compared, we tested all mice in a certain litter with the correct genotype. In all behavioral experiments, the experimenter was blind either to the genotype of the animals or to their treatment.

Chemogenic experiments followed a cross-over design. Comparisons were made between the effects of CNO or vehicle treatment. In experiments involving tetanus toxin-mediated silencing of neurons, cre negative littermates were used as controls with the experimenter being blinded to the genotype. Comparisons were made before and 5–7 days after the RVM injection of AAVs containing the coding sequence for TetLC using a 2-Way ANOVA to test for a genotype * time interaction.

### Virus injections

The dorsal horn of the lumbar spinal cord was injected as previously described[70,71]. Mice were anesthetized with 5% isoflurane which was maintained at 1–3% through a face mask during all surgeries. Analgesia (buprenorphine subcutaneously injected at 0.1–0.2 mg/kg) was provided before and after each surgery. During surgery, body temperature was maintained using a heated mat, and vitamin A cream was applied to protect the eyes. The skin above the injection site was shaved and disinfected with betadine. An incision was made to expose the T13 vertebra. The surrounding tissues were separated from the vertebral column and the T13 vertebra was clamped in position using spinal adapters. A borehole was made in the middle of the T13 vertebra on one side and AAVs were injected into the spinal dorsal horn at a depth of 200–300 μm below the spinal surface approximately 400 μm lateral to the central artery. In some anatomical tracing experiments, both sides of the spinal cord were injected in a similar manner. For virus injections, 3 × 300 nl virus solution was injected along the rostrocaudal extent of the spinal cord at an infusion rate of 50 nl/min. The viral vectors used in this study are listed in the Supplementary Table 3.

For injections into the RVM, the head was fixed in position using adjustable cheek bars. Injection coordinates were determined by the location of the retrogradely labeled RVM neurons that were identified in initial experiments, which relative to bregma were, and −5.8, ±0.5, 5.9 (rostrocaudal, mediolateral, dorsoventral respectively). Viruses were infused at 50–100 nl/min to a total volume of 1 μl for each injection. Injections were made using a motorized frame controlled by Neurostar Stereodrive software. The skull was installed in the frame to be as straight and as flat as possible (difference in z position between lambda and bregma <100 μm). To compensate for variations in tilt and scaling, adjustments to the injection target were made in the software relative to four positions on the surface of the skull (Bregma, Lambda, 2 mm to the right of the midline, and 2 mm to the left of the midline).

### Tissue preparation and immunohistochemistry

Mice were transcardially perfused with freshly prepared 4% paraformaldehyde (PFA) (room temperature, dissolved in 0.1 M phosphate buffer, adjusted to pH 7.4) following perfusion with 0.1 M phosphate buffer. Nervous tissues were quickly dissected and post-fixed in 4% PFA at 4°C for 2 h. For CLARITY tissue clearing experiments, tissues were post-fixed for 24 h at 4°C after dissection (see below for more details). After post-fixation, brains or spinal cords were placed in 30% sucrose solution (dissolved in 0.1 M phosphate buffer) for 24–72 h for cryoprotection of the tissue. These were then rinsed with 0.1 M phosphate buffer before being embedded in NEG-50 mounting medium. Tissues were sliced at 60 μm on a sliding blade microtome (Hyrax KS 34, histocam AG) and stored as free-floating sections. For long-term storage, free-floating sections were stored in antifreeze medium (50 mM) sodium phosphate buffer, 30% ethylene glycol, 15% glucose, and sodium azide (200 mg/l) at −20°C until required.

For immunostaining, sections were first washed three times in 0.1 M phosphate buffer and incubated in 50% EtOH for 30 min at room temperature, followed by three rinses in PBS with added salt (8 g NaCl per liter of PBS). Sections were incubated in a mixture of primary antibodies for two nights at 4°C. The antibodies were reconstituted in a solution containing PBS with added salt, 0.03% Triton-X, and 10% normal donkey serum. All primary antibodies and dilutions used in this study are listed in the Supplementary Table 1. Tissue sections were rinsed three times in PBS with added salt before being incubated in species specific secondary antibodies overnight at 4°C (see Supplementary Table 1). Finally, tissue sections were rinsed three times in PBS with added salt and mounted on microscope slides with Dako anti-fade medium.

### Tissue clearing and light sheet microscopy

Following a 24-h post-fixation in 4% PFA at 4°C, tissues were placed in a pre-chilled hydrogel solution (containing 45 ml PBS, 5 ml 40% polyacrylamide, and 125 mg VA-044) sealed in an air tight container for >2 days at 4°C. Samples were baked at 37°C for 3 h to allow polymerization of the hydrogel. Tissues were then transferred to an SDS-based clearing solution (containing 24.7 g boric acid, 80 g Sodium dodecyl sulfate in 2 l ddH$_2$O, adjusted to pH 8.5), and lipids were removed by active electrophoretic clearing for 3–5 h[72]. The clearing solution was circulated and chilled throughout the tissue clearing to avoid SDS warming and precipitation. Following active clearing, tissues were rinsed twice in PBS-T (containing 0.1% Triton-X) and then placed in histodenz solution overnight to allow for refractive index matching (refractive index adjusted to 1.4655 with PBS). Samples were glued to a platform and placed in an imaging cuvette filled with index-matched histodenz before being installed in a MesoSPIM light sheet microscope[27]. Images were acquired using MesoSPIM control and processed with ImageJ/FIJI.

### Multiplex fluorescent in situ hybridization (mFISH)

For fluorescent multiplex in situ hybridization experiments, brains were quickly dissected following euthanasia (<5 min). The hindbrains were isolated from the forebrain and cerebellum before being rapidly frozen in liquid nitrogen. Hindbrain tissues were embedded in Neg50 mounting media which was frozen for cryostat sectioning.

Tissue blocks were installed in a cryostat (Hyrax 60, histocam AG), and sections were taken at 20 μm thickness which were mounted directly onto superfrost microscope slices. Slides were stored at −80°C before FISH experiments. All in situ hybridization experiments followed the guidelines according to the RNAscope® Fluorescent Multiplex in situ hybridization v1 kit. Tissues were fixed for 15 min at 4°C in freshly prepared 4% PFA dissolved in PBS. Following fixation, the slides were dehydrated through an alcohol series (50%, 70%, 100% and 100% again), for 5 min at each EtOH concentration. The tissue containing region of the slide was delineated with a heat-resistant fat pen and allowed to dry. Tissues were treated with a Protease solution (Protease IV) for 30 min at room temperature before hybridization. Protease was removed and slides were briefly rinsed twice in PBS, Z probes (ACD bio) were hybridized to the tissues for 2 h in a 40°C purpose build

oven. Probes were removed and slides were rinsed twice in wash buffer for 2 min each before and between all amplification stages. Amplification of the hybridized probes were achieved using the amplification reagents 1, 2, 3, and finally were either labeled with 4A, 4B, or 4C (for 30, 15, 30, and 15 min respectively) in a 40°C oven. Slides were counterstained with DAPI for 10 min and mounted in DAKO antifade mounting medium. The probes used in this study are listed in the Supplementary Table 2.

Cells were counted using a CellProfiler pipeline to identify nuclei with DAPI staining, and to assign fluorescent puncta from each detected mRNA species within a small area surrounding the nuclei. Cells were counted as positive if they contained above the 90th percentile of puncta/nuclei for each fluorophore, determined by running a 3-plex negative control on an adjacent slide in parallel with the same amplification reagents[73]. Alternatively, experiments performed using the v2 version of the RNAscope® Fluorescent Multiplex in situ hybridization kit, cells were counted manually using FIJI with the cell counter plugin.

## Image acquisition and analysis
For an overview of injection sites and imaging entire brain sections, epifluorescence images were acquired using a Zeiss Axio Scan.Z1 slide scanner. For quantification of retrogradely labeled cells in the RVM, image stacks were acquired at 5 μm z-spacing using a Zeiss LSM 800 confocal microscope. Confocal scans were made using 488, 561 and 640 nm lasers and the pinhole was set to 1 Airy Unit for reliable optical sectioning. Image stacks were acquired where immunoreactivity of all antigens was clearly visible. Acquisition settings were kept the same within each quantification experiment and acquired images were analyzed offline with FIJI using the cell counter plugin. Data were processed in Microsoft Excel and were presented and analyzed with GraphPad Prism 8.

## TRAP2 labeling of activated neurons
To label projection neurons in the RVM that were activated by noxious stimuli, TRAP2 animals were used either in combination with an Ai65 reporter line, or by using an intersectional viral strategy. For the noxious stimulus, capsaicin (20 μl, 1 mg/ml) was injected into the plantar surface of the right forepaw[32]. After 1 h, 4 hydroxytamoxifen (4-OHT Sigma-Aldrich H6278) (1 mg/ml) was injected i.p. (10 mg/kg) for entry of the cre-ER into the nucleus to permit cre-dependent recombination. Stock solutions of 4-OHT were prepared at 40 mg/ml in warm DMSO (50°C) and were kept in aliquots at −20°C. To dissolve 4-OHT in an aqueous solution, saline containing 2% Tween 80 and 2.5% DMSO, and was warmed to 50°C during preparation of the working solution[74]. Experiments were performed at least 10 days following the final stimulation paired with 4-OHT. One or two stimulation 4-OHT pairings were performed to efficiently label as many pain-activated neurons as possible.

## Behavioral assays
For activation or silencing of descending RVM projections, we used an intersectional strategy in which an AAV2retro.flex.FLPo virus (also containing either a BFP or mCherry coding sequence) was used to transduce descending cre-expressing neurons. One week later, mice received RVM injections of AAVs containing a FLPo-dependent effector (either hM3Dq or TetLC). For DREADD experiments, 10 days incubation time was given to allow the expression of the receptor, whereas 5–7 days were given for TetLC expression before sensory and motor testing. Before experiments mice were acclimatized to the behavioral setup for at least 1 h. For the Hargreaves, cold plantar, electronic von Frey, and Rotarod assays, an average from six measurements was recorded per time point. For brush and pin prick tests, an average from 10 response scores was recorded per time point. All measurements were taken from both hindlimbs of all animals, and in some experiments, the forepaws were also tested, using the same scoring criteria for testing hindpaws.

For behavioral testing of TRAP2 mice, three experimental groups were used. (i) pre-TRAP, TRAP2 animals that received injections depicted in Fig. 8D, before the capsaicin forepaw injection and/or 4-OHT delivery. (ii) post-TRAP, animals that received the injections shown in Fig. 8D at least 10 days after forepaw capsaicin injection and 4-OHT. (iii) control-TRAP, animals that received the injections in Fig. 8D, 10 days after 4-OHT delivery but with no forepaw capsaicin injection.

## Hargreaves test
To measure heat sensitivity, paws were stimulated with an infrared heat source (Hargreaves plantar assay IITC). Mice were placed on a transparent platform preheated to 30°C, and withdrawal latencies were recorded in response to heating from an infrared heat source using an inbuilt timer. The stimulation intensity was set to 20% heater power for each test, and a cutoff time of 32 s was used to prevent potential tissue damage, with 3–5 min between stimulating the same paw to avoid tissue sensitization.

## Cold plantar assay
Mice were placed on a 5 mm borosilicate glass platform and were stimulated from beneath with dry ice pellets. The time taken between touching the glass from below to the withdrawal of the paw was measured and 20 s was used as the cutoff time to avoid any tissue damage. The same paw was only tested again after a 3–5 min recovery time.

## Electronic von Frey
Mechanical thresholds were determined using an electronic von Frey algesiometer (IITC). Animals were adapted on a mesh surface, and the plantar surface of each paw was stimulated with a bendable plastic filament attached to a pressure sensitive probe. Pressure was applied to the plantar surface gradually at a constant speed until the animal withdrew its paw. The maximum pressure that was present when the animal withdrew its paw was recorded.

## Dynamic allodynia (brush) test
Sensitivity to dynamic mechanical stimuli was a measured by gently brushing the hindpaw plantar surface with a paintbrush at a constant speed from heel to toe. Response scores were given for each stimulation and were based on methods previously described, with some adaptations to the scoring[75,76]. In brief, animals that did not withdraw their paw were given a score of 0, and animals that quickly withdrew were given a score of 1. Animals that withdrew and kept the paw raised (extended lifting) for >0.5 s were given a score of 2, and animals that flinched or licked the paw post-stimulation were scored as 3. Repeated flinching and licking of the stimulated paw were given a score of 4.

## Manual von Frey test
Von Frey thresholds were determined using a set of calibrated von Frey filaments and a simplified version of the up-down method. In brief, the glabrous skin of the plantar surface of the paw was stimulated for 3–5 s starting with the 0.4 g filament. Gentle pressure was applied until the filament started to bend, and each filament was used up to 5 times. A positive response was recorded when the mouse withdrew or flinched its paw, and a negative response was a lack of paw withdrawal. If three responses were positive the next lighter filament was chosen, and if there were three negative responses the next higher weight filament was used. This process was repeated five times, and the 50% withdrawal threshold was calculated for each paw.

## Rotarod

Sensorimotor coordination was evaluated using an accelerating rotarod, and the time taken for animals to fall from the rotating barrel was recorded. The barrel rotated from 4 to 40 rpm accelerating at a constant rate over a period of 300 s. Values were not included if the animal jumped from the barrel in the direction of rotation, and if the animal jumped in >50% of trials for a given time point these data were discarded.

## CFA inflammation

To induce inflammatory hyperalgesia, CFA (Sigma Aldrich F5881) was injected into the plantar surface of the ipsilateral hindpaw. Under anesthesia (5% isoflurane for induction, 1–2% for maintenance) 20 μl of CFA was injected subcutaneously into the plantar surface of the paw. The needle was left in place for 5–10 s after injection to reduce backflow. Mice were tested for up to 2 days following the injection.

## Chronic constriction injury model

Neuropathic pain was induced by applying a CCI to the left sciatic nerve proximal to its trifurcation. Three loose ligatures were put around the sciatic nerve, using 5-0 surgical silk, while mice were anesthetized with isoflurane 1–3%. The skin was closed with 2–3 sutures. Mice were tested on day 7 after CCI surgery.

## Capsaicin eye assay

Mice were adapted to a cylindrical plexiglass chamber for 15 min. Animals were restrained and 10 μl capsaicin (0.1% in 5% Tween 5% ethanol PBS) or vehicle was dropped onto the eye. Mice were returned to the chamber and the number of forepaw wipes directed to the affected eye were counted within the following 3 min.

## Chemogenetic experiments

Stock solutions of CNO were stored at a 500x concentration in DMSO at 100 mg/ml. Immediately prior to injection, CNO was diluted in sterile filtered saline. For chemgenetic experiments, CNO was injected either i.p. (2 mg/kg) or intrathecally (at 1 μg in 10 μl). In i.p. injection experiments, animals were tested directly before and 1–3 h after i.p. injections. For intrathecal injection, animals had the hair above the lumbar spinal segments shaved and removed with hair removal cream at least 1 day before behavioral experiments. Mice were briefly anesthetized with 2–3% isoflurane (<2 min) and placed on a circular falcon tube to expose the intervertebral spaces. The T13 vertebra was identified, secured, and 10 μl CNO (1 μg in 10 μl) was injected intrathecally with a 29 G insulin syringe. Injection sites were confirmed by the presence of a tail flick reflex during injection. Animals were returned to the testing arena, and sensory testing was performed at least 15 min after animals regained consciousness.

## In vivo optogenetic stimulation for behavioral experiments

Eight to 10 week-old vGATcre mice were implanted with spinal cannulas for local light delivery as described previously[77]. Implantations were made 2 weeks after brain injection of 1 μl of AAV9.-flex.hChR2_EYFP bilaterally in the LPGi. In brief, mice were anesthetized with 5% isoflurane and kept under anesthesia with 2–3% isoflurane on a stereotaxic frame throughout the duration of the surgery. After shaving the fur on the back of the mouse, a skin incision was made to expose the vertebral column and cuts were performed on the muscles located medially to the longitudinal tendons running along the vertebral bodies, to better expose the vertebrae. The vertebral column was subsequently clamped at the level of the thirteenth thoracic vertebra (T13). After carefully removing connective tissue and absorbing excessive bleeding with collagen strips (Lyostypt, B. Braun), a hole was drilled approximately 1 mm left of the spinous process of the T13 vertebra. After placing the cannulas (Doric lenses) with ferrules (⌀ 1.25 mm) on the drilled hole, two layers of 3 M™ Scotchbond™

Universal Self Etch Adhesive were applied and cured with UV light for 10 s each to fix the cannula in place. To ensure a more stable implant, Tetric Evoflow® A1 (Ivoclar vivadent) was subsequently applied and also cured with UV light for 10 s. The muscles along the vertebral column were then sutured using absorbable sutures (Safil 5-0, B. Braun). The skin was sutured with non-absorbable threads (Dafilon 6-0, B. Braun). Mice were allowed to recover on a heat pad. Behavioral experiments were conducted 10 days after surgery.

To measure optogenetically-evoked behavior, mice were placed in plexiglass cylinders (10 × 10 cm wide, 25 cm high). Immediately before sensory testing, cannulas were connected through a mating sleeve to the optic fiber (0.39 NA multimode fiber-optic patch cable, Thorlabs, Inc) with a rotatory joint to allow free movement of the mouse. The fiber was coupled to a blue LED module (PlexBright blue LED module, 470 nm, Plexon, Inc) and intensity was controlled through a current generator (Plexon, LED Driver LD-1, Plexon, Inc). An output current of 50 mA was sufficient to produce a light power ranging between $0.2 \pm 0.06$ mW (irradiance = 1.72 mW/mm²) at the outlet of the implanted cannula. Light power was measured ex vivo after the end of the sensory testing, using a fiber optic power meters with internal sensor (Thorlabs, Inc). The sensitivity to heat and cold was tested using the Hargreaves and Cold Plantar assay, respectively, before stimulation (baseline) and during stimulation (20 Hz burst for 1 s, every 10 s). Both tests were run with a 1-day interval in between.

## Slice preparation, optogenetics and whole cell recording

Spinal cord slices were prepared for electrophysiology experiments in a similar manner to previous studies[11]. Mice were decapitated while under isoflurane anesthesia, and the vertebral column was rapidly dissected and placed in oxygenated ice-cold dissection solution (containing in mM: 65 NaCl, 105 sucrose, 2.5 KCl, 1.25 NaH$_2$PO$_4$, 25 NaHCO$_3$, 25 glucose, 0.5 CaCl$_2$, 7 MgCl$_2$). The back of the animal was pinned to the bottom of a Sylgard-coated plate with the ventral surface facing up, and a laminectomy was performed to expose the spinal cord. The spinal cord was then carefully removed from the vertebral column and the lumbar and cervical enlargements were isolated. The ventral side of each spinal cord enlargement was then glued to an agar block, mounted in a slicing chamber and installed in a vibrating blade microtome (D.S.K microslicer DTK1000). Transverse slices of 300–350 μm thickness were taken and transferred to a holding chamber filled with oxygenated artificial cerebrospinal fluid (aCSF) (containing in mM: 120 NaCl, 26 NaHCO$_3$, 1.25 NaH$_2$PO$_4$, 2.5 KCl, 5 HEPES, 14.6 glucose, 2 CaCl$_2$, 1 MgCl$_2$; pH 7.35–7.40, 305–315 mOsm) warmed to 34°C. Slices were left to recover for at least 30 min before recordings were taken.

For the preparation of hindbrain slices, brains were quickly removed following euthanasia of the animal. The brain was placed in ice cold dissection solution, and the hindbrain was isolated by removing the forebrain and cerebellum. The hindbrain was glued to an agar block and transverse sections were cut at 250 μm, which were held in an incubation chamber filled with warm aCSF (34°C) until recordings were taken.

Targeted whole-cell recordings were taken at room temperature using a HEKA EPC10 amplifier with Patchmaster software (HEKA Elektronik) at a sampling frequency of 20 kHz. For optogenetic experiments, cells in the spinal dorsal horn (laminae I–III) were randomly targeted using DIC optics. A Cs-based internal solution was used (containing in mM: 120 CsCl, 10 HEPES, 0.05 EGTA, 2 MgCl$_2$, 2 Mg-ATP, 0.1 Na-GTP, 5 QX-314) and recordings were taken in voltage clamp mode, with blue light stimuli (4 ms, 473 nm) delivered through the objective lens at a frequency of 0.1 Hz from a monochromator (TILL photonics). To confirm the inhibitory nature of light evoked synaptic events, bicuculline (20 μM) and strychnine (0.5 μM) were bath applied to the slice during recording.

For targeted recordings of RVM projection neurons, the spinal cords of vGAT[cre] animals were injected with AAV2retro.flex.tdTOM or AAV2retro.flex.eGFP 1 week before the preparation of hindbrain slices. Labeled cells were visualized by epifluorescence illumination using a monochromator and were targeted for whole-cell recording. Recordings were taken in current clamp mode using a potassium gluconate internal solution (containing in mM, 130 K-Gluconate, 5 NaCl, 1 EGTA, 10 HEPES, 5 Mg-ATP, 0.5 Na-GTP, 2 biocytin) to determine spontaneous activity. Following recording, the recording electrode was slowly retracted, and the slice was fixed overnight at 4°C in 4% PFA. Fixed hindbrain slices were processed to reveal biocytin-labeled cells and immunoreacted to confirm their tdTOM or eGFP content.

Access resistance was monitored throughout each experiment, and traces from each experiment were analyzed further if the access resistance changed < 30% during the recording. For light-evoked responses displayed in Fig. 4, averaged traces from 10 consecutive stimuli are shown in black, and individual traces used to generate the averaged trace are shown in gray. Blue lines indicate the time of blue-light stimulation, and all traces are aligned to this.

### Spinal cord in vivo electrophysiology
Extracellular single-unit recordings were made from neurons in the superficial dorsal horn (100–300 μm from the dorsal surface) as described previously[78]. In brief, mice were anesthetized with 2% isoflurane and maintained at 37°C body temperature. After T13-L2 laminectomy, the lumbar segments were exposed, and the animal was mounted in a stereotaxic frame (David Kopf Instruments). A glass insulated carbon filament electrode (4–6 MΩ) was used for recording. Neurons were identified by monitoring background activity and responses to innocuous and noxious search stimuli. Neurons were selected that responded more strongly to noxious than innocuous stimuli, corresponding to the WDR type. For optogenetic activation of the superficial dorsal horn axon terminals of vGAT RVM[SC] neurons, AAV9.hEF1a.Dio.ChR2 was injected stereotaxically into the RVM of 6-week old vGAT[cre] mice (control mice were injected with rAAV9.E-F1a.Dio.mcherry). Recordings were made 3–4 weeks after vectors injections. Neuronal activity (spikes/s) was measured for 1–3 min in the absence of intentional stimulation (background activity) and during innocuous and noxious mechanical test stimulation. To assess the impact of vGAT RVM[SC] neuron activity on the responses of the recorded neuron, dorsal horn axon terminals of vGAT RVM[SC] neurons were stimulated with blue light (473 nm, 2 min, 20 Hz, 5–10 mW) using an optical fiber placed close to the recording electrode. Background and evoked activity of neurons was recorded before (baseline), during ("ON"), and after ("OFF") blue light stimulation.

### Statistical tests
Normal distribution was tested for all data sets using the Shapiro-Wilk test. If the data set followed a normal distribution, unpaired or paired t tests, or ANOVA were used. If the data set did not pass, non-parametric tests were used to assess statistical significance. $p < 0.05$ was considered statistically significant.

### Data collection, storage, and presentation
Data were analyzed and presented using GraphPad Prism 8. Figures were arranged in Affinity Designer 2.

### Reporting summary
Further information on research design is available in the Nature Portfolio Reporting Summary linked to this article.

## Data availability
All unique materials are available from the authors. All quantitative data generated in this study are provided in the Source Data file. Raw data acquired in these experiments has been uploaded to https://doi. org/10.5281/zenodo.18983374 and is available for download. Source data are provided with this paper.

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

## Acknowledgements

The authors thank the Center for Microscopy and Image Analysis at the University of Zurich for assistance with the tissue clearing and light-sheet microscopy experiments, and Louis Scheurer and Eva Roth for technical assistance. The work was supported by grants from the Swiss National Science Foundation (grant number: 310030_197888 to H.U.Z.), a grant from the European Union's Horizon 2020 research and innovation actions (agreement no. 101016787 to H.U.Z.), the Clinical Research Priority Program 'Pain—from phenotypes to mechanism' of the Faculty of Medicine, University of Zurich, to H.U.Z., and was supported by the intramural research program of the National Institute of Dental and Craniofacial Research, National Institutes of Health, project ZIADE000721-19 (M.A.H.).

## Author contributions

R.P.G., M.S. H.W. and H.U.Z. conceived and designed the study. R.P.G., M.S., C.B. and F.P. performed behavioral assays, R.P.G., M.S., K.W., s.d'A. and T.A. performed anatomical tracing and histology experiments. R.P.G. and M.R. performed whole-cell electrophysiology experiments, G.J. and V.N. designed and performed in vivo electrophysiological recordings. M.H., P.S. and M.A.H. provided input on the interpretation of the data and discussion of the results. R.P.G., M.S., H.W. and H.U.Z. wrote the manuscript with inputs from all authors.

## Competing interests

The authors declare no competing interests.
