## [Transparent Peer Review file · Nature Communications]

Descending Inhibitory Rostral Ventromedial Medulla Neurons Cause Widespread Antinociception and Contribute to the Pain-Inhibits-Pain Phenomenon

Corresponding Author: Professor Hanns Ulrich Zeilhofer

Version 0:

Reviewer comments:

Reviewer #1

(Remarks to the Author)

In this report, authors using optogenetic methods to re-investigate descending inhibition of pain from the RVM to the spinal cord. Although it is great to see similar findings as previous reports (see Reviews from Sandkuhler, Gebhart et al), the study lacks of novel mechanisms for such modulation. Only behavioral responses were used for the evaluation of nociceptive responses, additional electrophysiological recordings from dorsal horn neurons combined with optogenetic stimulation may strength the argument.

Major:

1. Authors failed to acknowledge previous reports of biphasic (facilitatory and inhibitory) descending transmission from this regions (see Zhuo and Gebhart for spinal electrophysiology; Porreca for chronic pain). I wonder it may be caused by the limitation of techniques. How intensity-dependent stimulation may play out in light stimulation? does virus affect the healthy status of RVM neurons?
2. Authors need to mention ON- off- cells reported by Fields et al for RVM descending modulation. I wonder if they are belonged to the types of cells in the RVM.
3. Additional recordings from dorsal horn neurons, including their responses to peripheral thermal/mechanical stimuli are needed to confirm the inhibition at sensory transmission. Some control experiments from vspinal motor neurons would be helpful too.
4. Authors need to cite old literature from Wills, Gebhart, Sandkuhler labs that consistently demonstrate that descending inhibitory modulation from the RVM are bilateral to spinal cord dorsal horn neurons. It is good to have virus methods to confirm this, however, it alone is not novel.

Reviewer #2

(Remarks to the Author)

The manuscript by Ganley et al uses viral genetic tools for tracing and behavioral assessment of brainstem neurons that influence nociceptive thresholds, mechanical and/or thermal in mice.

1. There have been several studies of how the RVM impacts nociception, and thus the authors should put into context better which findings reported in this manuscript are novel to the field and why. It should be very clear to the broad audience of Nature Communications what information was lacking and what is now provided. There are only vague sentences in intro and discussion. Both the intro and discussion need to be re-written for a broader audience to include more in-depth explanations.
2. The authors show that back-labeling bilaterally from the lumbar dorsal horn using an intersectional strategy with VGAT Cre and retro AAV captures GABAergic neurons in the RVM including in the NRM, Gi, LPGi. Using two different colored reporters for the back-labeling of ipsi and contra, the authors show that a significant proportion of RVM neurons project bilaterally. This is true whether both injections are at lumbar area or one is lumbar and the other cervical – hence widespread innervation along the cord by single RVM neurons.
3. The study also highlights that these GABAergic RVM neurons also project to supraspinal areas, which is a confound for

the study since the spinal projections are not specifically manipulated. The authors should provide a more compelling and detailed explanation for why these circuits are not responsible for their observations. It is dismissed only with a brief and nondescript sentence in the discussion. If a compelling explanation cannot be provided, experimental evidence that supports a dependence specifically on the spinal projections will be needed.

4. hM3Dq -mediated activation of GABAergic RVM neurons from lumbar ipsilateral injection. Impact on von Frey, heat, cold but not pinprick or brush. Should discuss why not pinprick or brush. With respect to brush, inhibition of the neurons with TetLC impacts brush.

5. hM3Dq -mediated activation of the GABAergic RVM neurons labeled from lumbar unilateral injection impacts the Hargreaves (heat sensitivity) across all four paws equally but the effect on cold sensitivity occurs most profoundly at the ipsi hind paw – this should be discussed.

6. TetLC mediated inhibition of GABAergic RVM neurons labeled from lumbar unilateral injection increases sensitivity to brush, von Frey but no impact on heat or cold. There is some concern that the expression of TetLC is not as widespread or robust (indirectly looking at GFP) compared to the hM3Dq indirectly looking at mCherry. Also, no image of the dorsal horn GFP expression in the TetLC mice. An experiment that demonstrates the TetLC is working well is needed here. For example, co-inject the TetLC and the hM3Dq and show that the impact of CNO-mediated activation of the RVM neurons on heat and cold is blocked.

7. Taken together, the results as shown indicate activating RVM GABA neurons can suppress heat, cold, and von Frey and inhibiting RVM GABA neurons can increase brush and von Frey. In this sense, only von Frey sensitivity seems to be bidirectionally modulated.

8. Slice recordings of the RVM GABA neurons show some are tonically active. The discussion of these neurons in relation to the behavior could be explained in more detail- provide the logic or rationale for why tonically active neurons are modulating mechanical sensitivity in a more localized fashion (i.e. inhibiting them only impacted mechanical and only at hind paw).

9. The impact of hM3Dq activation of GABAergic RVM neurons on CFA von Frey and CCI von Frey were different. CFA von Frey showed much greater recovery -going over baseline levels. CCI did not recover to baseline. Heat sensitivity with CFA recovered. The impact between CFA and CCI is therefore not the same and CCI has inflammatory and neuropathic components so perhaps CCI recovery reflects an impact on inflammatory component only.

10. Show Fos induction in GABA RVM neurons due to injection of capsaicin in forepaw – shown in Fig. 7. In Figure 8, FosTrap2 mice are used which allows excitatory and inhibitory neurons in RVM to be captured with cap forepaw injection. With n=2, most neurons are GABAergic 80 vs 20%. Captured with hM3Dq. Do these neurons project to other brain areas or only to the spinal cord? Number of neurons is not reported but looks sparse and limited to LPGi. Reactivation of these neurons increases heat and cold thresholds but has no impact on von Frey or brush.

The authors should make it clear if the identification of LPGi neurons as mediators of CPM or DNIC is novel. The authors in the discussion mention it as a feed-forward circuit but do not explain why it is feed forward rather than feedback. The paragraph on DNIC with respect to the identified circuit and the method of activation is not clear. For example, the authors mention it is activated by noxious stimuli at a distant site but what does distant site mean?

Other comments

Figure 1A – missing v for VGLUT2 on graph

Figure 2C left side- the labeling of brain regions in the slices is unclear.

Figure 3 should say that behavior is measured at ipsilateral hind paw in figure legend.

Figure 4B- the way the authors show the impact of strychnine and bicuculine on the Chr2 evoked currents is unclear. I assume the currents in gray are polysynaptic and that the flat also gray line under the currents is what was observed in the presence of antagonists but this is normally shown above the currents lining up with the light activation (blue line).

Figure 4F- the labels are cutoff.

Typo Page 10 line 3.

Reviewer #3

(Remarks to the Author)

The manuscript by Ganley et al is an intersectional study investigating the basis of RVM mediated inhibition of superficial dorsal horn (DH) physiology and thus descending inhibition of nociception. This is an important area of study in pain research as it is one of key cornerstones of clinically utilised analgesia, yet the fundamental properties of this process is not understood. The MS is well written and in large part the experiments are well designed and well presented. I do have a few concerns at this stage which I will outline below.

My first concern is the use of male mice exclusively in this study. There is a growing literature that pain processing differs between the sexes and that this is the case in PAG mediated analgesia, much of which relies on spinal projections from the RVM. The authors fail to investigate whether sex has an impact on their findings. Ideally this would be performed but would require considerable time, effort and expense. At a minimum some discussion of sex-based differences is required and an acknowledgement that the findings of this study are may not be applicable to females.

Now to the second major concern. Was the C-FOS used as a proxy for identification of neuronal activity? Had the authors thought of performing RVM recordings to assess whether ON, OFF or neutral cells were impacted by the CNO treatment? This point then reflects my main concern with many of the outcomes from slice-based electrophysiology. Surely an in vivo approach to spinal and RVM recordings would have enabled many of the same data to be produced and allowed these to be made in an intact physiological context? Further it would have allowed for a clarification/classification of the neurons in the RVM manipulated and therefore permitted comparison from physiological and pharmacological studies in the literature. This is a key deficit in the field that the neurochemistry of physiological defined RVM neurons which respond to noxious stimuli is less than completely understood.

Many of the points raised by the authors rely on extrapolations rather than direct measures. In terms of DNIC (and clearly not CPM) mechanistic insight is gained via whole animal, in vivo, electrophysiological and behavioural measures. Whether DNIC can be perturbed by activation/inhibition of RVM neurons requires an intact prep. Is it possible for a small set of studies to be conducted to confirm these ex vivo data from this study? This is especially important for the studies reflected in figure 4 which investigate functional synapses, yet use slices with isolated spinal cord in addition to the whole animal behaviour.

Panel 3B and C seems a little out of focus (this may be my PDF) the lumbar injection site images are a little too low powered to enable a good inspection of superficial DH.

The title for the legend of figure 6 implies only a reversal of inflammatory pain yet CCI mediated hyperalgesia is partially reversed too, a change would help the reader.

The methods are unclear in some elements of experimental design. How were group sizes (power) calculated? How were animals assigned to treatment groups? Blinding is mentioned with regards to CNO treatment but there are other interventions in these studies in which experimenters should have been blind to- were they? There are a great many studies described in this study, some more detail on statistical treatment and the justification for those tests is required. For example was normality testing of all data sets performed?

Version 1:

Reviewer comments:

Reviewer #1

(Remarks to the Author)

Authors have performed some additional experiments to address my concern on spinal nociceptive transmission. However, these experiments were only performed in normal mice but not chronic pain ones. It is well known that spinal nociceptive transmission is significantly enhanced after periphery injuries. In previous studies, many investigators reported that such inhibition was diminished in animal models of chronic pain. Maybe it will be good for authors to repeat such experiments using classic electrical stimulation, in order to compare with optogenetic methods. Secondly, descending facilitation from RVM to spinal cord has been well documented, it seems to be missing using optogenetic methods. Again, authors may need to repeat previous studies using old methods.

Reviewer #2

(Remarks to the Author)

With a few exceptions below, the authors addressed the earlier concerns. The work makes a significant contribution to the field.

Comments:

1. "...neuropathic diseases." There is a typo but also it is unclear that neuropathic component of pain was affected as stated the earlier review. Could have been only inflammatory component.
2. It is not conclusive as described to say that the presence of an effect in one location (shutting down WDR lamina I neurons) is causal if it is unclear whether another effect is simultaneously happening in another location (modulation of

supraspinal circuit). The physiological effect on spinal cord has to be isolated so impact on the spinal cord can be monitored. Behavior has to be measured in this same condition.

3. Related to this, the following statement is too vague. "Furthermore, previous work has 333 shown that inhibitory effects of RVM stimulation required intact dorsolateral funiculi, 334 indicating that descending projections to the spinal cord are required for these effects 39, 40, 41, 335 42."

I can look up the references but can the authors just state what the reported inhibitory effects are that require descending projections?

Version 2:

Reviewer comments:

Reviewer #2

(Remarks to the Author)

The authors have addressed remaining concerns.

Point-by-point reply

Reviewer 1:

In this report, authors using optogenetic methods to re-investigate descending inhibition of pain from the RVM to the spina cord. Although it is great to see similar findings as previous reports (see Reviews from Sandkuhler, Gebhart et al), the study lacks of novel mechanisms for such modulation. Only behavioral responses were used for the evaluation of nociceptive responses, additional electrophysiological recordings from dorsal horn neurons combined with optogenetic stimulation may strength the argument.

Major:

1. Authors failed to acknowledge previous reports of biphasic (facilitatory and inhibitory) descending transmission from this regions (see Zhuo and Gebhart for spinal electrophysiology; Porreca for chronic pain). I wonder it may be caused by the limitation of techniques. How intensity-dependent stimulation may play out in light stimulation? does virus affect the healthy status of RVM neurons?

We thank the reviewer for this advice. We now discuss previous work by the Gebhart and Porreca labs on biphasic descending pain control by the RVM (p.10). In our study, we have used adenoassociated viruses (AAVs) for gene delivery. AAVs are well tolerated by neurons and even considered for gene therapy in human patients (e.g. PMID 40076831). Our functional (electrophysiological and chemogenetics) and morphological analyses of AAV transfected neurons failed to reveal any signs of neuronal toxicity. We attribute the selective inhibitory effect of chemogenetic activation of vGAT RVM^{SC} neurons to the specificity of our approach rather than to neurotoxic effects of AAVs.

2. Authors need to mention ON- off- cells reported by Fields et al for RVM descending modulation. I wonder if they are belonged to the types of cells in the RVM.

We thank the reviewer also for this comment. We discuss ON, OFF and NEURTAL RVM cells in the revised manuscript (p.10). The descending vGAT RVM neurons studied in the present manuscript share many characteristics with OFF cells.

3. Additional recordings from dorsal horn neurons, including their responses to peripheral thermal/mechanical stimuli are needed to confirm the inhibition at sensory transmission. Some control experiments from vspinal motor neurons would be helpful too.

We agree with the reviewer that *in vivo* recordings from dorsal horn WDR neurons would strengthen our study. Therefore, we initiated a collaboration with the Neugebauer lab at the Texas Tech University Health Science Center. Consistent with the effects of vGAT RVM^{SC} neurons occurring in the spinal cord, the new experiments show that optogenetic stimulation of descending vGAT RVM^{SC} virtually abolishes the responses of dorsal horn WDR neurons to noxious stimulation (revised figure 3 and extended data figure 3 and p.6).

4. Authors need to cite old literature from Wills, Gebhart, Sandkuhler labs that consistently demonstrate that descending inhibitory modulation from the RVM are bilateral to spinal cord dorsal horn neurons. It is good to have virus methods to confirm this, however, it alone is not novel.

Previous work e.g. by the Sandkuhler lab from 1995 and 1996 (PMIDs 7675181 and 8931107) used kainate injections into the ventromedial medulla (targeted onto the nucleus raphe magnus (NRM) to activated RVM neurons. Subsequent activation of spinal neurons was detected as c-fos expression. c-fos induction occurred bilateral and along the rostrocaudal axis. Although very informative at that time, with the techniques available it was not possible to (a) selectively activate descending RVM neuron, (b) restrict the activation to a certain neurochemically defined neuron type, and (c) to detect inhibition of spinal cord neurons. Our new study therefore significantly advances previous knowledge. Our new tools allowed us to specifically activate genetically defined subsets of RVM neurons, to restrict activation to spinally projecting neurons and to assess the effects of not only excitation but also inhibition. Our intersectional

approaches unequivocally demonstrate for the first time that individual vGAT RVM^{SC} neurons traced from the lumbar spinal cord innervate both sides of the spinal cord along its rostrocaudal axis.

Reviewer #2 (Remarks to the Author):

The manuscript by Ganley et al uses viral genetic tools for tracing and behavioral assessment of brainstem neurons that influence nociceptive thresholds, mechanical and/or thermal in mice.

1. There have been several studies of how the RVM impacts nociception, and thus the authors should put into context better which findings reported in this manuscript are novel to the field and why. It should be very clear to the broad audience of Nature Communications what information was lacking and what is now provided. There are only vague sentences in intro and discussion. Both the intro and discussion need to be re-written for a broader audience to include more in-depth explanations.

We have re-written major parts of the discussion and now put our finding in the context of previous work on descending pain modulation by the RVM and on earlier studies addressing DNIC and CPM (see also our responses to reviewer 1, points 1 and 4). We state more clearly what was already known and what our study adds through the use of state-of-the-art neuroscience tools, such as viral tracing, and intersectional genetics based chemogenetics.

2. The authors show that back-labeling bilaterally from the lumbar dorsal horn using an intersectional strategy with VGAT Cre and retro AAV captures GABAergic neurons in the RVM including in the NRM, Gi, LPGi. Using two different colored reporters for the back-labeling of ipsi and contra, the authors show that a significant proportion of RVM neurons project bilaterally. This is true whether both injections are at lumbar area or one is lumbar and the other cervical – hence widespread innervation along the cord by single RVM neurons.

We thank the reviewer for this encouraging statement.

3. The study also highlights that these GABAergic RVM neurons also project to supraspinal areas, which is a confound for the study since the spinal projections are not specifically manipulated. The authors should provide a more compelling and detailed explanation for why these circuits are not responsible for their observations. It is dismissed only with a brief and nondescript sentence in the discussion. If a compelling explanation cannot be provided, experimental evidence that supports a dependence specifically on the spinal projections will be needed.

The reviewer points to an important aspect of our study. We agree that the work cited in the previous version of our manuscript may not be considered conclusive enough to exclude an antinociceptive effect via RVM input occurring through supraspinal sites. Therefore, in our revised manuscript we added *in vivo* electrophysiological experiments in collaboration with the Neugebauer lab at the Texas Tech University Health Science Center. These experiments show that optogenetic stimulation of the spinal terminals of vGAT RVM^{SC} neurons is sufficient to virtually block activation of dorsal horn wide dynamic range neurons by nociceptive input.

4. hM3Dq-mediated activation of GABAergic RVM neurons from lumbar ipsilateral injection. Impact on von Frey, heat, cold but not pinprick or brush. Should discuss why not pinprick or brush. With respect to brush, inhibition of the neurons with TetLC impacts brush.

The differential impact of chemogenetic activation and tetanus toxin-mediated silencing on thermal versus mechanical sensitivity may suggest the presence of different populations of vGAT RVM^{SC} neurons targeting different dorsal horn neuron populations as discussed in the manuscript (p.10, last paragraph). Without a detailed knowledge of the spinal target neurons of the vGAT RVM^{SC} neuron populations, we can only speculate about the underlying mechanisms.

5. hM3Dq -mediated activation of the GABAergic RVM neurons labeled from lumbar unilateral injection impacts the Hargreaves (heat sensitivity) across all four paws equally but the effect on cold sensitivity occurs most profoundly at the ipsi hind paw – this should be discussed.

We agree with the reviewer that this is an interesting finding. As discussed above, at present, we could only speculate about the underlying circuits.

6. TetLC mediated inhibition of GABAergic RVM neurons labeled from lumbar unilateral injection increases sensitivity to brush, von Frey but no impact on heat or cold. There is some concern that the expression of TetLC is not as widespread or robust (indirectly looking at GFP) compared to the hM3Dq indirectly looking at mCherry. Also, no image of the dorsal horn GFP expression in the TetLC mice. An experiment that demonstrates the TetLC is working well is needed here. For example, co-inject the TetLC and the hM3Dq and show that the impact of CNO-mediated activation of the RVM neurons on heat and cold is blocked.

Tetanus toxin light chain (TetLC)-mediated silencing is usually much more potent than chemogenetic approaches. We now provide a high-resolution image of the of the injected area (Fig.5C) which illustrates that virtually all retrogradely labeled (magenta) neurons also express TetLC (green), with basically no magenta-only neurons visible in the panel of Fig. 5C.

7. Taken together, the results as shown indicate activating RVM GABA neurons can suppress heat, cold, and von Frey and inhibiting RVM GABA neurons can increase brush and von Frey. In this sense, only von Frey sensitivity seems to be bidirectionally modulated.

We agree with the reviewer.

8. Slice recordings of the RVM GABA neurons show some are tonically active. The discussion of these neurons in relation to the behavior could be explained in more detail- provide the logic or rationale for why tonically active neurons are modulating mechanical sensitivity in a more localized fashion (i.e. inhibiting them only impacted mechanical and only at hind paw).

We thank the reviewer for comment. We discuss a possible heterogeneity of vGAT RVM^{SC} neurons on page 10.

9. The impact of hM3Dq activation of GABAergic RVM neurons on CFA von Frey and CCI von Frey were different. CFA von Frey showed much greater recovery -going over baseline levels. CCI did not recover to baseline. Heat sensitivity with CFA recovered. The impact between CFA and CCI is therefore not the same and CCI has inflammatory and neuropathic components so perhaps CCI recovery reflects an impact on inflammatory component only.

The contribution of inflammation to various nerve injury models is an interesting question. It is well established that macrophages and cytokines contribute to nerve injury-mediated sensitization (e.g., PMID 20003309; PMID 27387067). In our previous experiments, we have shown that a pathway that is critically involved in inflammatory hyperalgesia does not contribute to sensitization in the CCI model (PMID 16846696). We feel that the present wording is appropriate, as, in the abstract, we speak of chronic pain states, and, in the results section, we specifically define the model used in our experiments.

10. Show Fos induction in GABA RVM neurons due to injection of capsaicin in forepaw – shown in Fig. 7. In Figure 8, FosTrap2 mice are used which allows excitatory and inhibitory neurons in RVM to be captured with cap forepaw injection. With n=2, most neurons are GABAergic 80 vs 20%. Captured with hM3Dq. Do these neurons project to other brain areas or only to the spinal cord? Number of neurons is not reported but looks sparse and limited to LPGi. Reactivation of these neurons increases heat and cold thresholds but has no impact on von Frey or brush.

The TRAP captured neurons project to similar supraspinal sites (lateral parabrachial nucleus and PAG) as the neurons labeled from the spinal cord.

The authors should make it clear if the identification of LPGi neurons as mediators of CPM or DNIC is novel. While the RVM (or more specifically the NRM) has already previously been proposed as a CNS area involved in DNIC, we are not aware of studies that specifically identify a contribution of neurons in the LPGi.

The authors in the discussion mention it as a feed-forward circuit but do not explain why it is feed forward rather than feedback.

We have removed this statement, as, depending on the viewpoint, it may also be considered feedback.

The paragraph on DNIC with respect to the identified circuit and the method of activation is not clear.

We have re-written the discussion on DNIC and CPM (p.11) and hope that it is now clearer.

For example, the authors mention it is activated by noxious stimuli at a distant site but what does distant site mean?

We use the term distant to explain that the test stimulus and the conditioning stimulus were applied on different sides (left versus right) and at different levels (e.g., forepaw versus hindpaw). We hope that this is made clear in the revised manuscript.

Previous work has suggested an involvement of OFF cells in the RVM as mediators of DNIC nucleus raphe magnus

We thank the reviewer for this comment and quote the paper by Chebbi and colleagues (PMID 24681000).

Other comments

Figure 1A – missing v for VGLUT2

has been corrected

on graph Figure 2C left side- the labeling of brain regions in the slices is unclear.

has been corrected

Figure 3 should say that behavior is measured at ipsilateral hind paw in figure legend.

has been corrected.

Figure 4B- the way the authors show the impact of strychnine and bicuculine on the ChR2 evoked currents is unclear. I assume the currents in gray are polysynaptic and that the flat also gray line under the currents is what was observed in the presence of antagonists but this is normally shown above the currents lining up with the light activation (blue line).

we have re-arranged Fig. 4B

Figure 4F- the labels are cutoff. Typo Page 10 line 3.
has been corrected.

Reviewer #3 (Remarks to the Author):

The manuscript by Ganley et al is an intersectional study investigating the basis of RVM mediated inhibition of superficial dorsal horn (DH) physiology and thus descending inhibition of nociception. This is an important area of study in pain research as it is one of key cornerstones of clinically utilised analgesia, yet the fundamental properties of this process is not understood. The MS is well written and in large part the experiments are well designed and well presented. I do have a few concerns at this stage which I will outline below.

My first concern is the use of male mice exclusively in this study. There is a growing literature that pain processing differs between the sexes and that this is the case in PAG mediated analgesia, much of which relies on spinal projections from the RVM. The authors fail to investigate whether sex has an impact on their findings. Ideally this would be performed but would require considerable time, effort and expense. At a minimum some discussion of sex-based differences is required and an acknowledgement that the findings of this study are may not be applicable to females.

In the present study we have used mice of both sexes for most experiments. In the revised figure we indicate the sex of the mice using different symbols (filled versus open circles for male and female mice, respectively). Some of the morphological experiments and the *in vivo* electrophysiology were done in male mice only. This is also indicated in the figures.

Now to the second major concern. Was the C-FOS used as a proxy for identification of neuronal activity? Had the authors thought of performing RVM recordings to assess whether ON, OFF or neutral cells were impacted by the CNO treatment?

c-fos labeling was used as a marker of neuronal activation. We agree that *in vivo* recordings of RVM neurons ideally performed in behaving mice would provide more information, especially because of the temporal resolution, which is completely lacking in case of c-fos stainings. However, such recordings are extremely challenging. They require surgery, implantations of electrodes, optical fibers or lenses. Given that the LPGi is a very small structure at a delicate site of the brain stem, it is likely that such manipulations would have interfered with the integrity of the circuit. We have however performed *in vivo* recordings from spinal wide dynamic range neurons in the spinal cord (Fig. 3 E,F and page 6).

This point then reflects my main concern with many of the outcomes from slice-based electrophysiology. Surely an *in vivo* approach to spinal and RVM recordings would have enabled many of the same data to be produced and allowed these to be made in an intact physiological context? Further it would have allowed for a clarification/classification of the neurons in the RVM manipulated and therefore permitted comparison from physiological and pharmacological studies in the literature. This is a key deficit in the field that the neurochemistry of physiological defined RVM neurons which respond to noxious stimuli is less than completely understood.

We certainly agree with reviewer that *in vivo* recordings from descending RVM neurons would add to our study. However, as pointed out above, we cannot currently overcome the many challenging technical requirements of such experiments. We discuss some work on the neurochemical characteristics of ON, OFF and NEURTRAL RVM cells (p. 10).

Many of the points raised by the authors rely on extrapolations rather than direct measures. In terms of DNIC (and clearly not CPM) mechanistic insight is gained via whole animal, *in vivo*, electrophysiological and behavioural measures. Whether DNIC can be perturbed by activation/inhibition of RVM neurons requires an intact prep. Is it possible for a small set of studies to be conducted to confirm these *ex vivo*

data from this study? This is especially important for the studies reflected in figure 4 which investigate functional synapses, yet use slices with isolated spinal cord in addition to the whole animal behaviour.

We appreciate the reviewer's suggestion and collaborated with the Neugebauer lab at the Texas Tech University Health Science Center to perform electrophysiological recordings from dorsal horn neurons in an intact preparation. We optogenetically stimulated the spinal terminals of vGAT RVM^{SC} neurons and monitored the impact of this stimulation on the activity of dorsal horn wide dynamic range neurons evoked by noxious and innocuous stimulation of their receptive field. These experiments revealed that optogenetic stimulation virtually abolished stimulus evoked action potential firing in these neurons.

Panel 3B and C seems a little out of focus (this may be my PDF) the lumbar injection site images are a little too low powered to enable a good inspection of superficial DH.

We now show the images a higher resolution.

The title for the legend of figure 6 implies only a reversal of inflammatory pain yet CCI mediated hyperalgesia is partially reversed too, a change would help the reader.

Has been changed.

The methods are unclear in some elements of experimental design. How were group sizes (power) calculated?

As our project would be classified as "exploratory", we did not perform sample size calculations but rather estimated required group sizes from previously conducted similar experiments by our group.

How were animals assigned to treatment groups?

Whenever possible, we used a random number generator and the identification numbers of the mice in our animal registration system to attribute mice to the different treatment groups. This was not possible when different genotypes were compared. In these cases, we tested all mice in a certain litter with the correct genotype.

Blinding is mentioned with regards to CNO treatment but there are other interventions in these studies in which experimenters should have been blind to- were they?

In all behavioral experiments, the experimenter was blind either to the genotype of the animals or to their treatment.

There are a great many studies described in this study, some more detail on statistical treatment and the justification for those tests is required. For example was normality testing of all data sets performed?

We have tested all data sets for normal distribution using the Shapiro-Wilk test. When the data set did not pass this test, a non-parametric test was used to analyze for statistical significance. We have also added more details on the outcomes of the statistical tests.

Point-by-point reply

Reviewer #1 (Remarks to the Author):

Authors have performed some additional experiments to address my concern on spinal nociceptive transmission. However, these experiments were only performed in normal mice but not chronic pain ones.

We have carefully discussed the possibility of adding additional *in vivo* WDR neuron recordings from diseased mice, which were notably not request in the first round of reviews. We are not convinced that these experiments would substantially strengthen the manuscript. Our existing *in vivo* WDR recordings already indicate that the antihyperalgesic effects observed in behavioral pain tests are associated with reduced action potential firing in dorsal horn WDR neurons. We discuss this in our revised manuscript (p. 10 / l. 7-12). In addition, we have performed behavioral experiments in both inflamed and neuropathic mice that clearly demonstrate antihyperalgesic effects following activation of descending RVM vGAT neurons. Based on these data, we currently have no reason to expect fundamentally different outcomes in mice with pathological pain states.

It is well known that spinal nociceptive transmission is significantly enhanced after periphery injuries. In previous studies, many investigators reported that such inhibition was diminished in animal models of chronic pain.

We fully accept the reviewer's comments. We have ourselves contributed to the identification of mechanism that enhance spinal nociception in inflammatory and neuropathic pain states.

Maybe it will be good for authors to repeat such experiments using classic electrical stimulation, in order to compare with optogenetic methods.

It is not unexpected that experiments employing classical, non-specific electrical or chemical stimulation of all RVM neurons and genetics-based approaches that selectively recruit the spinally projecting inhibitory RVM neurons yield different results. We feel that repeating these early experiments using non-specific methods would not be meaningful when newer, more precise techniques are available. Instead, we will acknowledge this point in our Discussion (p. 10 / l. 36 – p. 11 / l. 2).

Secondly, descending facilitation from RVM to spinal cord has been well documented, it seems to be missing using optogenetic methods. Again, authors may need to repeat previous studies using old methods.

We agree with the reviewer that stimulation of the RVM can have opposing effects on spinal nociception. We now discuss the heterogeneity of responses in the revised manuscript (p. 10 / l. 37 – p. 11, l. 3 and p. 11 / l. 29-30). Facilitating responses have in fact been reported also after optogenetic stimulation (François et al., *Neuron* 2017). As already discussed, we attribute these differences to different subnuclei targeted in the different studies (p. 11 / l. 20 – 28).

Reviewer #2 (Remarks to the Author):

With a few exceptions below, the authors addressed the earlier concerns. The work makes a significant contribution to the field.

Comments:

1. "...neuropathic diseases." There is a typo but also it is unclear that neuropathic component of pain was affected as stated the earlier review. Could have been only inflammatory component.

We have corrected the typo (thank you for the careful reading). We now discuss that the relatively small antihyperalgesic effect of vGAT RVM^{SC} neuron stimulation in the CCI model might potentially be due to its restriction to the inflammatory component of CCI induced hyperalgesia (p.12/l.1 - 6). We also adapted the wording throughout the manuscript and speak of "hyperalgesic disease states" or "pathological pain states" instead of neuropathic pain to take this possibility into account.

2. It is not conclusive as described to say that the presence of an effect in one location (shutting down WDR lamina I neurons) is causal if it is unclear whether another effect is simultaneously happening in another location (modulation of supraspinal circuit). The physiological effect on spinal cord has to be isolated so impact on the spinal cord can be monitored. Behavior has to be measured in this same condition.

We appreciate the reviewer's critique and have performed additional behavioral experiments with local spinal chemogenetic or optogenetic stimulation (revised extended data figure 3 A-F, and p. 6/l. 5-15, and p. 10/l. 4-10).

3. Related to this, the following statement is too vague. "Furthermore, previous work has 333 shown that inhibitory effects of RVM stimulation required intact dorsolateral funiculi, 334 indicating that descending projections to the spinal cord are required for these effects 39, 40, 41, 335 42."

I can look up the references but can the authors just state what the reported inhibitory effects are that require descending projections?

We now explicitly mention the experiments described in the previous publication (p.10/l. 4-7)

Point-by-point reply

Reviewer #2 (Remarks to the Author):

The authors have addressed remaining concerns.

Thank you for this positive comment.